# A Cross-Sectional Study on the Psychological Effects of Different Exercises on Elderly Women

**DOI:** 10.3390/brainsci15090918

**Published:** 2025-08-26

**Authors:** Wen Qi, Hongli Yu, Dominika Wilczyńska

**Affiliations:** 1Department of Physical Culture, Gdansk University of Physical Education and Sport, 80-336 Gdansk, Poland; qi.wen@awf.gda.pl; 2College of Physical Education, Sichuan University of Science and Engineering, Zigong 643000, China; hongli.yu@suse.edu.cn; 3Faculty of Social and Humanities, WSB Merito University, 80-266 Gdansk, Poland

**Keywords:** elderly women, mental health, sleep quality, oriental exercises, cross-sectional study

## Abstract

**Background:** Exercise has been widely recognized as an effective non-pharmacological intervention method for maintaining the mental health and sleep quality of the elderly. However, existing studies rarely distinguish the types of exercise based on cultural attributes, and this is even more so for the situation of elderly women. This oversight limits the accuracy of exercise recommendations for this population. Hence, this study aimed to compare the differential effects of three types of exercise—Oriental exercise, general exercise, and sedentary leisure exercise (SLE)—on physiological indicators and mental health in elderly women and to examine the associations between physiological indicators, sleep quality, and mental health. **Methods**: A total of 386 participants were included in the study, comprising 252 individuals in the Oriental exercise group (mean ± SD: 67.83 ± 5.36), 110 individuals in the general exercise group (mean ± SD: 67.19 ± 4.47), and 24 individuals in the SLE group (mean ± SD: 67.38 ± 4.75). Psychological tests (Pittsburgh Sleep Quality Index (PSQI), the Satisfaction with Life Scale (SWLS), the Subjective Well-being Scale (SWB), the Geriatric Depression Scale (GDS), the Geriatric Anxiety Scale-20 (GAI-20), and the Exercise Motivation Scale (SMS)) and physiological measurements (such as body weight and pulmonary capacity) were conducted. ANOVA was performed to compare the effects of the three types of exercise on physiological and psychological outcomes, and correlation analyses were conducted to evaluate the associations between different types of exercise and mental health indicators, physiological indicators, and sleep quality. **Results**: Primary Objective: One-way ANOVA showed significant group differences in key outcomes. Bonferroni post hoc tests confirmed the following: Oriental exercise > general exercise in SWB (*p* = 0.032); general exercise > SLE in sleep quality (PSQI, *p* = 0.028) and lower body weight (*p* = 0.019); Oriental exercise > general exercise in vital capacity (*p* < 0.001). Secondary Objective: Pearson correlations revealed the following: PSQI negatively correlated with life satisfaction (SWLS: *p* = 0.035) and positively with GDS (*p* = 0.021); SWB positively correlated with SWLS (*p* < 0.001) and negatively with GDS (*p* = 0.002). **Conclusions**: Different exercise modalities have distinct benefits for elderly women. Oriental exercise improved vital capacity and well-being, while general exercise reduced body weight and enhanced sleep quality compared with sedentary leisure activity. Tailored exercise programs may effectively promote both physical and mental health in this population.

## 1. Introduction

Elderly women experience unique mental health and sleep-related challenges that substantially influence their overall quality of life. Depressive symptoms and poor sleep quality are particularly prevalent, with epidemiological evidence indicating that women aged 60–74 are at higher risk of depression than men, driven by factors such as hormonal changes, caregiving responsibilities, and social isolation [1]. More than half the women in this age group experience sleep disturbances, which can exacerbate depressive symptoms and reduce subjective well-being—a key mental health indicator in the present study [2]. In the context of global population aging, developing targeted interventions to improve sleep quality, alleviate depressive symptoms, and enhance overall well-being is a pressing public health priority.

Exercise is widely recognized as an effective non-pharmacological strategy for improving sleep quality, mental health outcomes, and physiological parameters such as body weight and lung capacity in elderly women. Evidence supports the benefits of aerobic training, resistance exercise, and combined mind–body practices such as Baduanjin and Tai Chi [3,4]. However, existing studies seldom differentiate exercise types based on cultural attributes, despite the potential for culturally tailored approaches to optimize both adherence and effectiveness.

Culturally adapted interventions, particularly those rooted in traditional practices, may enhance participation and therapeutic outcomes by fostering familiarity, cultural identity, and intrinsic motivation [5]. Oriental exercises such as Baduanjin and Tai Chi integrate physical movement with mindfulness and breath regulation, aligning with holistic health perspectives common in many non-Western societies [6,7,8,9]. In contrast, general exercises—classified according to exercise physiology—include aerobic activities (e.g., running, swimming), anaerobic strength training, and competitive sports, emphasizing quantitative physical improvements such as cardiopulmonary endurance and muscle mass [10,11,12]. These activities, not bound to a specific philosophical system, have broad global applicability and reflect the increasing westernization of sports participation among elderly women in urban China (e.g., modern dance, gym-based activities, and square dancing).

SLE represents a third category, encompassing activities such as fishing or chess. While these may meet certain definitional criteria for “sport” due to their structured and competitive elements [13], their energy expenditure is minimal [14]. In this study, SLE serves as a low-activity control condition, enabling a clearer contrast with physically active modalities and allowing examination of whether low-intensity activities confer benefits over complete sedentariness.

Henan Province was chosen as the study site due to its strong cultural traditions, high participation in Oriental exercises among elderly women, and rapidly aging population, particularly in rural areas [7]. This context offers a relevant setting for examining the interplay between culturally distinctive and more globally common exercise modalities.

Based on theoretical frameworks and preliminary evidence, the present study had three objectives: (1) to compare the effects of Oriental exercises, general exercises, and SLE on physiological and psychological outcomes in community-dwelling elderly women; (2) to explore correlations between physiological indicators, sleep quality, and mental health. We hypothesized that

**H1.** 
*Participants engaging in Oriental exercises would report significantly better mental health and sleep quality than those engaging in general exercises or SLE.*


**H2.** 
*Participants engaging in general exercises would report significantly better mental health and sleep quality than those in the SLE group.*


**H3.** 
*Mental health scores would be positively correlated with sleep quality scores across all participants, regardless of exercise group.*


## 2. Materials and Methods

### 2.1. Study Design

This study is a cross-sectional survey design to evaluate the effects of physical activities on the psychological health of elderly women. Data were collected via a questionnaire survey, capturing their demographic characteristics, physiological indicators, mental health status, and exercise habits. Exercise types were classified into three categories: Oriental exercise, general exercise, and SLE. This cross-sectional study adhered to the STROBE (Strengthening the Reporting of Observational Studies in Epidemiology) guidelines.

### 2.2. Participants

#### 2.2.1. Inclusion Criteria

Participants were community-dwelling women aged 60–74 years, classified by the World Health Organization (WHO) as “young-old”, a demographic with relatively preserved adaptive capacity and suitability for regular physical activity to attenuate age-related physiological decline [15]. Eligible participants were in generally good health, without serious cardiovascular (e.g., uncontrolled hypertension, heart failure) or respiratory diseases, and were capable of engaging in low- to moderate-intensity exercise. Additional criteria included the absence of cognitive impairment (e.g., dementia), a stable mental state, and no medical contraindications to exercise.

#### 2.2.2. Exclusion Criteria

Individuals were excluded if they had severe comorbidities or functional limitations (e.g., advanced heart or lung disease, severe musculoskeletal disorders, or severe mental illness) that precluded completion of the study protocol. Participants were also excluded if they did not meet the minimum physical activity threshold of three sessions per week, each lasting ≥ 30 min.

#### 2.2.3. Recruitment and Group Classification

During recruitment, participants were stratified by age (60–64, 65–69, 70–74 years) and by habitual activity type to ensure balanced representation across exercise categories. Initial classification was based on a structured baseline questionnaire, capturing self-reported exercise types, frequency (sessions/week), and average duration (minutes/session) over the preceding three months. All information was verified in face-to-face interviews by trained research assistants prior to any physiological or psychological assessment.

#### 2.2.4. Exercise Group Definitions

Participants were allocated to one of three mutually exclusive groups based on their habitual physical activity over the preceding three months. The Oriental exercise group comprised individuals performing Tai Chi, Baduanjin, or Qigong at least three times per week, with each session lasting ≥ 30 min [16]. The general exercise group included those regularly engaging (≥3 sessions/week) in aerobic activities, such as walking, cycling, or swimming, or in resistance training while undertaking Oriental exercise less than twice per week or for a total duration of <40 min per week [17]. The SLE group consisted of participants performing fewer than one session of moderate-intensity physical activity per week yet engaging in at least three leisure pursuits (e.g., card games, fishing) on a weekly basis, representing a low-energy-expenditure lifestyle for comparative purposes [18].

### 2.3. Sample Size

To ensure data representativeness and statistical significance, the sample size was calculated using the formula for cross-sectional studies. With a margin of error of ±5% and a 95% confidence level (*Z* = 1.96), the calculation was as follows:(1)$$n=\frac{Z^{2}\times p\times (1-p)}{E^{2}}$$
where *Z* is the *Z* value corresponding to the confidence level, which is 1.96 for a 95% confidence level; *p* is the estimate proportion, typically 0.5 for the most conservative estimate for maximum sample size; and *E* is the margin of error, set at 0.05 [19,20].

Substituting the values into the formula,(2)$$n=\frac{1.96^{2}\times 0.5\times (1-0.5)}{0.05^{2}}=384$$

Thus, a minimum of 384 participants was required. To account for potential low response rates and data loss, we targeted 500–600 elderly women to enhance reliability and statistical power [21]. The questionnaire’s overall Cronbach’s *α* coefficient was 0.81, indicating acceptable internal consistency [22]. Ultimately, 500 questionnaires were collected, with 386 deemed valid. Stratified sampling was used to ensure representation across age groups (60–64, 65–69, 70–74 years) and exercise types. The final sample size (*n* = 386) exceeded the minimum requirement (*n* = 384), ensuring sufficient statistical power for primary analyses (Figure 1).

This study was conducted in accordance with the recommendations and ethical guidelines of the ethics review committee (approval number: 2025-LW-D0220). All participants provided written informed consent in accordance with the Declaration of Helsinki.

To ensure the accuracy and representativeness of the study results, the following factors were considered in the sample size adjustment:

Cross-sectional surveys typically require a fixed sample size; increasing it improves statistical power, particularly for multi-group comparisons.

Anticipated invalid responses due to personal reasons or data omission were factored in; a larger sample size accommodates potential loss and ensures completeness.

As the study involves multiple subgroups (e.g., exercise types and health conditions), stratified sampling ensures adequate representation of subgroups, enhancing internal validity.

### 2.4. Procedures

This study aims to assess the health status, exercise habits, and participation in Oriental and non-Oriental physical activities in elderly women in Henan Province. The procedures were as follows.

#### 2.4.1. Data Collected

This cross-sectional observational study was conducted among elderly women recruited at community health centers and local elderly activity rooms from December 2024 to March 2025 in Henan Province, following the STROBE standard. Validated scales and physiological measurement methods were employed. Eligible participants were women aged between 60 and 74 years. After obtaining written informed consent, data collection was carried out in batches. First, the participants underwent eligibility screening and grouping. The participants (n = 386) were divided into three groups: the Oriental exercise group (n = 252, 67.83 ± 5.36 years), the general exercise group (n = 110, 67.19 ± 4.47 years), and the SLE group (n = 24, 67.38 ± 4.75 years) (Table 1). Then, the participants completed a structured questionnaire and underwent psychological tests using the PSQI, the SWLS, the SWB, the GDS, the GAI-20, and the SMS. Subsequently, physiological measurements (such as body weight and pulmonary capacity) were conducted.

Physiological indicators were assessed following standardized testing protocols. Body weight was measured to the nearest 0.1 kg using a calibrated electronic scale (Model HD-351, Tanita, Tokyo, Japan), with participants wearing light clothing and no shoes. Measurements were taken once after the participant had stood still for 3–5 s in the center of the scale platform. Vital capacity was determined using a portable spirometer (Model MicroQuark, COSMED, Rome, Italy) with participants standing upright. Each participant performed maximal inspiration followed by forceful and complete expiration. Three trials were conducted with at least 60 s of rest between attempts, and the highest value meeting the acceptability and repeatability standards of the American Thoracic Society/European Respiratory Society (ATS/ERS) was recorded for analysis. Resting heart rate was measured using a chest-strap heart rate monitor (Model CMS50DL, CONTEC, Qinhuangdao, China; accuracy ±1 bpm) in a quiet, temperature-controlled room (22–24 °C) after participants had been seated at rest for at least 5 min. Heartbeat was recorded continuously for 60 s, and the mean value was used for analysis; measurements were repeated if variation between consecutive readings exceeded 3 bpm.

All physiological measurements were performed by trained research staff, with instruments calibrated daily according to the manufacturers’ specifications. To measure the mental health status of the participants, in addition to collecting some basic demographic information, this study also used six classic psychological scales.

The sleep quality of the elderly participants in the most recent month was evaluated using PSQI. This index consists of 19 self-evaluated items and 5 other-evaluated items. Among them, the 19th self-evaluated item and 5 other-evaluated items are not included in the scoring, and 18 items constitute 7 components. Each component is scored on a scale of 0 to 3. The cumulative score of each component is the total PSQI score, which ranges from 0 to 21. The higher the score is, the worse the sleep quality. Cronbach’s α coefficient is 0.83 [23].

The subjective experience of individuals regarding their own quality of life was evaluated using SWLS. There were a total of five items, and a Likert 7-level scoring system was adopted (1 represents “completely inconsistent”, and 7 represents “completely consistent”), with a Cronbach alpha coefficient of 0.78 [24].

The SWB scale was used to assess an individual’s overall evaluation of their quality of life based on their own standards, including two major dimensions: emotion and cognition. The subjects’ feelings, reactions, and degrees of identification were taken as the assessment indicators, and a 6-point scale grading method was adopted. Its standard is “1”, indicating strong disagreement; “6” indicates strong agreement. The Cronbach alpha coefficient is 0.929 [25].

The use of the GDS can more sensitively examine the specific physical symptoms of elderly patients with depression, allowing the elderly to review their emotional states within a week and assess whether they have depression problems. There are 30 items, including the following symptoms: low mood, reduced activity, irritability, withdrawn and painful thoughts, and negative evaluations of the past, present, and future. Each item is a question, requiring the subjects to answer “Yes” or “No”. Among the 30 items, 10 were scored in reverse order (answering “No” indicates the existence of depression), and 20 were scored in positive order (answering “Yes” indicates the existence of depression). Each answer indicating depression is worth 1 point. A score of 0 to 10 indicates normal, and a score of 11 to 20 represents mild depression. Scores of 21 to 30 indicate moderate to severe depression. Cronbach’s *α* coefficient of the GDS is 0.793 [26].

The anxiety symptoms of the elderly participants were measured using GAI-20, covering anxiety manifestations in aspects such as psychology, cognition, and social interaction. The Likert 4-level scoring system (“0” indicating completely inconsistent, “3” indicating completely consistent) was used, with a total of 20 questions, and the Cronbach alpha coefficient was 0.91 [27].

The self-motivation ability or intrinsic motivation level of the elderly participants was measured using the SMS. This scale is divided into two dimensions, internal motivation and external motivation, with a total of 28 questions. It adopts the Likert 6-level evaluation system, where “1” indicates complete non-conformity and “6” indicates complete conformity. Cronbach’s *α* coefficient of the SMS is more than 0.60, indicating acceptable reliability [28].

#### 2.4.2. Data Analysis

All analyses were performed using SPSS version 22 (IBM Corp., Armonk, NY, USA), with descriptive statistics and inferential analyses reported according to standard epidemiological conventions. The matrix diagram for the relevant analysis was drawn using Origin 2021.

The measurement level of each variable was determined prior to analysis. Continuous variables were first tested for normality using the Shapiro–Wilk test and by visual inspection of histograms and Q–Q plots. Variables meeting normality and homogeneity of variance assumptions were analyzed with parametric methods; continuous variables were compared between groups using one-way ANOVA, and associations between them were examined using Pearson’s correlation coefficient [29].

Ordinal variables and continuous variables that did not meet normality assumptions were analyzed with non-parametric methods; group differences were tested using the Kruskal–Wallis H test, and associations were examined using Spearman’s rank correlation coefficient. This mixed approach allowed each variable to be analyzed using the most appropriate statistical method based on its measurement level and distribution. Statistical significance was set at *p* < 0.05 for all tests [30].

For descriptive statistical analysis, the mean ± standard deviation (SD) is used to represent it. Between-group comparisons of continuous outcomes (e.g., questionnaire scores, physiological measures) were performed using one-way ANOVA with Bonferroni-adjusted post hoc tests for multiple group comparisons, while independent samples *t*-tests were used for two-group contrasts (for example, urban vs. rural participants or different exercise intervention groups). Effect sizes were calculated as Cohen’s *d* for the *t*-test and Cohen’s *f* for ANOVA, with conventional benchmarks for interpretation (Cohen’s *d* 0.2 = small, 0.5 = medium, 0.8 = large; Cohen’s *f*: 0.1 = small, 0.25 = medium, 0.4 = large). Ninety-five percent confidence intervals (CIs) were computed for all estimated means and effect size metrics to convey the precision of the estimates. Associations between variables were assessed using Pearson’s correlation coefficient (r) for normally distributed data or Spearman’s rank-order correlation (ρ) for non-normally distributed data, with 95% confidence intervals calculated for each correlation coefficient. All *p*-values were adjusted for multiple comparisons using the Bonferroni method, and statistical significance was defined as *p* < 0.05 (two-tailed).

## 3. Results

This section provides a detailed description and statistical analysis of the collected data, focusing on the participants’ demographic characteristics, exercise habits, physical health status, and psychological health. The results are organized into four subsections for clarity.

### 3.1. Demographic Characteristics

Table 2 describes the demographic information of the participants. Numerical variables (such as age, height, weight, etc.) are presented in the form of “mean ± standard deviation (mean (SD))”. The data formats of each group are uniform, and the units are clearly labeled (such as yrs, cm, kg, etc.). Categorical variables (such as residence status, marital status, etc.) are presented as percentages (%). In terms of mental health, the test results of the Oriental group in the SWLS, SWB, GDS, and GAI-20 were superior to those of the other two groups.

### 3.2. The Influence of Different Types of Exercise on Mental Health Indicators of the Elderly

Table 3 presents the analysis of variance results of three groups of exercise patterns for different variables. The variance analysis of PSQI, SWB, GDS, vital capacity, and body weight between groups showed statistically significant differences, while there were no statistically significant differences in other variables. Subsequently, the Bonferroni method was used for the post hoc detection of PSQI, SWB, GDS, vital capacity, and body weight.

Through Bonferroni post hoc analysis of intergroup comparisons, it is known that in terms of physiological indices, there are significant differences in body weight between the general group and the SLE group, with the body weight of the general group being significantly lower than that of the SLE group (mean ± SD: 63.3 ± 6.33 vs. 66.20 ± 6.72, *p* = 0.014, *f* = 0.748); the vital capacity of the Oriental group is significantly higher than that of the general group (mean ± SD: 2047.08 ± 214.04 vs. 1916.11 ± 203.08, *p* = 0.001, *f* = 0.62). In terms of sleep quality and mental health, the sleep quality of the general group is significantly higher than that of the SLE group (mean ± SD: 8.23 ± 3.34 vs. 7.08 ± 3.65, *p* = 0.045, *f* = 0.343). SWB data indicate that there are significant differences between the Oriental group and the general group (mean ± SD: 80.61 ± 9.49 vs. 77.77 ± 8.85, *p* = 0.024, *f* = 0.341), and the SWB results of the Oriental group are significantly higher than those of the general group (mean ± SD: 80.61 ± 9.49 vs. 77.77 ± 8.85, *p* = 0.024, *f* = 0.341).

### 3.3. The Influence of Different Areas on the Mental Health of the Elderly

Independent sample *t*-tests were conducted to assess area disparities in six psychological constructs among elderly women: PSQI, SWLS, SWB, GDS, GAI-20, and SMS. As detailed in Table 4, no statistically significant area difference emerged for any outcome variable.

All analyses retained α = 0.05 and reported Cohen’s *d* effect sizes to quantify practical significance. These findings suggest area does not moderate sleep quality, life satisfaction, subjective well-being, or mental health symptoms in this population.

### 3.4. The Interrelationships Among Different Variables

This section explores the relationship between three types of exercise and different psychological indicators, physiological indicators, and sociodemographic factors, as well as a comprehensive perspective on how they affect the mental health and sleep quality of elderly women aged 60–74.

Figure 2 presents a correlation matrix, constructed using Origin software, that examines the relationships between indicators of psychological well-being and sleep quality. Each cell in the matrix displays the correlation coefficient between two variables, with circle size indicating the strength of the correlation and color representing the direction (red for positive, blue for negative).

As shown in Figure 2, VC was significantly positively correlated with SMS (*p* < 0.05); PSQI was significantly negatively correlated with SWLS and SWB (*p* < 0.05). Furthermore, PSQI was significantly positively correlated with GDS (*p* < 0.05); SWLS was significantly positively correlated with SWB (*p* < 0.05) and significantly negatively correlated with GDS (*p* < 0.05). SWB was significantly negatively correlated with GDS and GAI-20 (*p* < 0.05).

## 4. Discussion

Based on the study findings, all three initial hypotheses were partially supported. Consistent with H1, elderly women engaging in Oriental exercises demonstrated superior mental health and sleep-related outcomes to those performing general exercise or SLE, as evidenced by significantly higher SWB scores and greater vital capacity. H2 was also supported, as participants in the general exercise group exhibited significantly better sleep quality than those in the SLE group and lower body weight. Furthermore, correlational analyses provided support for H3, revealing that better sleep quality (lower PSQI scores) was associated with higher life satisfaction and SWB and fewer depressive symptoms. These results collectively underscore the beneficial role of both culturally specific and general exercise modalities in promoting psychological well-being and physiological health among elderly women, with Oriental exercise demonstrating the most pronounced advantages.

### 4.1. The Differences in Physiological and Psychological Mechanisms Between Different Types of Exercises

Our findings indicate clear differences in mental health outcomes between the three exercise modalities examined. Participants in the Oriental exercise group reported the most favorable mental health scores, followed by those in the general exercise group, with the SLE group scoring lowest.

One possible explanation for these differences is that Oriental exercises such as Tai Chi and Baduanjin integrate physical movement with mindfulness, breathing regulation, and social interaction [31]. These elements may enhance psychological benefits by reducing perceived stress, fostering emotional regulation, and increasing social connectedness—factors that have been positively associated with mental health in elderly women in prior observational studies. In contrast, general exercise, while beneficial for physical fitness and mood, may lack the same degree of mind–body integration or cultural resonance, which could partly explain the smaller effect size observed for mental health outcomes in this group [32].

SLE, although potentially cognitively stimulating, does not provide the physiological or psychosocial benefits associated with regular physical activity, which likely contributes to the less favorable mental health scores in this group. This gradient in outcomes is consistent with other cross-sectional research, which has found stronger mental health benefits from combined physical–cognitive–social exercise formats than from purely physical exercise or SLE [33].

### 4.2. The Mechanism of Sleep and Mental Health

This community-based cross-sectional study found that elderly women who frequently engaged in Oriental exercises performed better in terms of mental health and sleep quality than those who engaged in general exercise or SLE.

Our research results are largely consistent with those of several observational studies conducted on elderly women. Firstly, regarding the positive correlation between Oriental exercises and mental health (such as subjective well-being), our results are consistent with the results of a previous cross-sectional study conducted among elderly women in China [34,35].

Secondly, in this study, the general exercise group was observed to have better sleep quality and lower body weight than the SLE group, which is also supported by other observational studies [36,37]. Additionally, the Oriental exercise group showed better pulmonary function indicators (such as vital capacity), which has also been reported in other observational studies.

However, our results also indicate that the benefits shown by the Oriental exercise group in terms of mental health (such as life satisfaction) and sleep quality seem to be greater than the effects typically reported for ordinary aerobic exercises. This observation differs from the recent cross-sectional analysis results of Liu et al. [37], whose study found that Tai Chi practitioners and general exercisers did not have a significant difference in sleep quality [38]. This difference may be due to differences in study sample characteristics (such as regional cultural background), measurement tools, or specific definitions of activity categories. It is worth noting that the potential advantages of the Oriental exercises observed in this study may be related to the fact that they combine multiple factors such as physical activity, cognitive concentration, and potential social interaction. In the future, more rigorous designs, such as randomized controlled trials, will be needed to verify these associations [39].

### 4.3. Interrelationships Between Sleep Quality, Mental Health, and Physiological Function

Based on the observed correlations, our findings highlight a significant interrelationship between physiological, psychological, and sleep-related indicators among elderly women. VC was positively associated with SMS, suggesting that better pulmonary function may contribute to enhanced cognitive and psychological functioning in this population [40]. Conversely, sleep quality, as measured by the PSQI, demonstrated negative correlations with both SWLS and SWB and a positive correlation with GDS [41]. These results support the notion that poor sleep quality is closely linked to diminished mental health outcomes, aligning with previous research emphasizing the bidirectional relationship between sleep disturbances and depression in elderly women [42].

Furthermore, the positive association between SWLS and SWB, coupled with their negative correlation with GDS, underscores the interconnectedness of well-being and mental health among elderly women [43]. Notably, SWB was also negatively correlated with GAI-20, suggesting that higher subjective well-being is linked to lower psychological distress. Collectively, these findings confirm our third hypothesis (H3) regarding the positive relationship between mental health and sleep quality and emphasize the potential of interventions targeting both physiological fitness and sleep hygiene to improve overall mental health in aging populations [44]. These correlations provide a rationale for exercise interventions, particularly those that enhance cardiorespiratory function and promote restorative sleep, as a strategy to maintain and improve psychological well-being among elderly women.

### 4.4. Limitations and Future Directions

This study has three key limitations that need to be addressed. First, it is important to interpret the findings in light of the unequal group sizes, particularly the relatively small sample in the SLE group (n = 24). Smaller group sizes reduce statistical power and limit the precision of estimates for that group. Consequently, comparisons involving the SLE group should be viewed with caution, as the observed differences may underestimate or overestimate the true effects. Despite this limitation, the consistency of the observed trends with prior observational studies lends some support to the robustness of our findings. Second, as a cross-sectional study, we can only report associations between exercise types and outcomes (e.g., Oriental exercise and higher SWB) but cannot confirm causal relationships. For example, we cannot determine whether Oriental exercises cause better well-being or if individuals with higher well-being are more likely to choose Oriental exercises. Third, data were collected exclusively in Henan Province, where 65.3% of participants prefer Oriental exercises. This cultural homogeneity may limit the generalizability of findings to regions with different exercise traditions (e.g., urban areas with westernized sports trends).

Future research should focus on four directions: 1. expanding the SLE sample via stratified sampling and applying rigorous statistical models to enhance generalizability; 2. using wearable devices to monitor real-time physiological changes during exercise for deeper insights into Oriental exercises’ benefits; 3. conducting multicenter randomized controlled trials to assess long-term effects of Oriental exercises and develop standardized community programs; 4. conducting neuroendocrine studies to explore Oriental exercises’ regulation of cortisol and the autonomic nervous system.

## 5. Conclusions

This study demonstrates that exercise type significantly influences both physiological and mental health outcomes in elderly women. General exercises were associated with lower body weight and better sleep quality than SLE, while Oriental exercises yielded superior vital capacity and higher subjective well-being compared with general exercises. Correlation analyses revealed strong links between better sleep quality, higher life satisfaction, enhanced well-being, and reduced psychological distress. These findings highlight the potential of tailored exercise programs—particularly Oriental exercises—to promote integrated physical and psychological health in elderly women.

## Figures and Tables

**Figure 1 brainsci-15-00918-f001:**
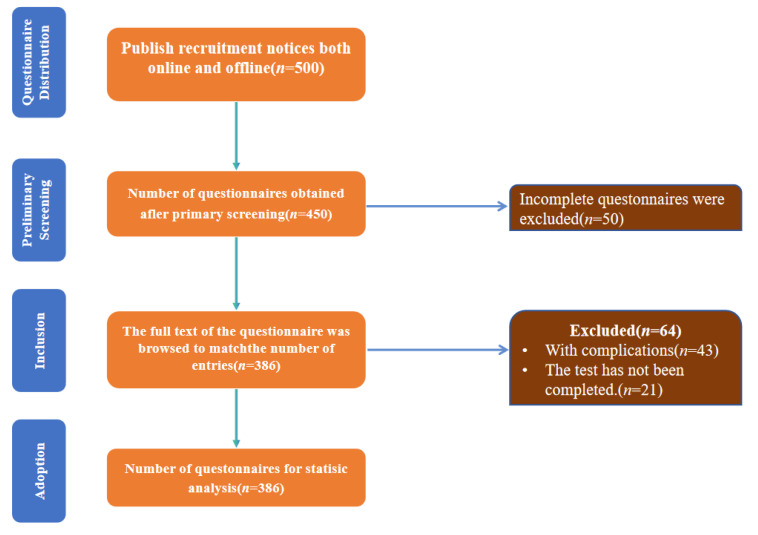
Flowchart of participant selection.

**Figure 2 brainsci-15-00918-f002:**
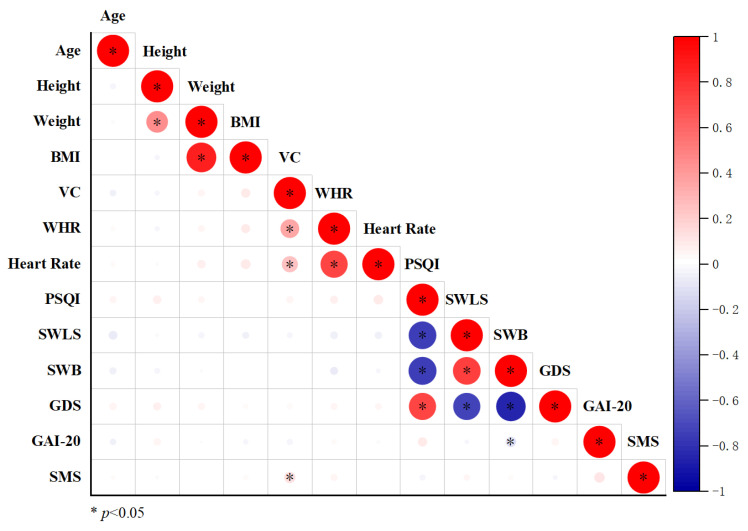
Correlation analysis of physiological indicators, sleep quality, and psychological variables in the elderly participants. Body Mass Index: BMI, Vital Capacity: VC, Waist–Hip Rate: WHR, Pittsburgh Sleep Quality Index: PSQI, Satisfaction with Life Scale: SWLS, Subjective Well-Being: SWB, Geriatric Depression Scale: GDS, Geriatric Anxiety Inventory-20: GAI-20, and Sport Motivation Scale: SMS.

**Table 1 brainsci-15-00918-t001:** Demographic and characteristics of participants by group.

Group	Oriental (*n* = 252)	General (*n* = 110)	SLE (*n* = 24)
Characteristic			
**Residence, %**			
City	136 (53.97)	36 (32.73)	12 (50.00)
Rural–Urban	116 (46.03)	74 (67.27)	12 (50.00)
**Marital status, %**			
Married	162 (64.29)	67 (60.91)	20 (83.33)
Widowed	53 (21.03)	27 (24.55)	3 (12.50)
Divorce	30 (11.90)	12 (10.90)	1 (4.17)
Spinsterhood	7 (2.78)	4 (3.64)	0 (0.00)
**Ethnic, %**			
Han	235 (93.25)	102 (92.73)	22 (91.67)
Minority	17 (6.75)	8 (7.27)	2 (8.33)
**Education, %**			
Primary School	63 (25.00)	89 (80.90)	5 (20.83)
Junior High School	145 (57.54)	15 (13.64)	14 (58.33)
High School	26 (10.32)	4 (3.64)	2 (8.33)
University	18 (7.14)	2 (1.82)	3 (12.51)
**Type of work, %**			
Manual Labor	200 (79.37)	100 (90.91)	18 (75.00)
Mental Labor	52 (20.63)	10 (9.09)	6 (25.00)
**Income, %**			
<CNY 3000	158 (62.70)	82 (74.55)	13 (54.17)
CNY 3001–5999	66 (26.19)	23 (20.90)	8 (33.33)
≥CNY 6000	28 (11.11)	5 (4.55)	3 (12.50)
**Exercise frequency (per week), %**			
1–2 times	170 (67.46)	62 (56.36)	20 (83.33)
3–4 times	64 (25.40)	35 (31.82)	4 (16.67)
More than 5 times	18 (7.14)	13 (11.82)	0 (0.00)
**Duration of exercise, %**			
<30 min	45 (17.86)	20 (18.19)	2 (8.33)
30–60 min	119 (47.22)	62 (56.36)	13 (54.17)
61–120 min	61 (24.21)	17 (15.45)	4 (16.67)
>120 min	27 (10.71)	11 (10.00)	5 (20.83)

Oriental exercise group: Oriental; General exercise group: General; Sedentary leisure exercise group: SLE.

**Table 2 brainsci-15-00918-t002:** The demographic characteristics and mental health status of the participants.

Group	Oriental (*n* = 252)	General (*n* = 110)	SLE (*n* = 24)
Variable			
**Mean age (SD), yrs**	67.83 ± 5.36	67.19 ± 4.47	67.38 ± 4.75
**Mean height (SD), cm**	166.4 ± 5.0	166.55 ± 5.72	167.63 ± 4.92
**Mean weigh (SD), kg**	64.16 ± 6.39	63.3 ± 6.33	66.20 ± 6.72
**Mean BMI (SD), kg/m^2^**	23.15 ± 1.90	23.00 ± 1.89	23.50 ± 2.20
**Vital Capacity (mL)**	2047.08 ± 214.04	1916.11 ± 203.08	2021.13 ± 199.34
**Waistline (cm)**	84.49 ± 8.96	84.38 ± 8.99	86.21 ± 10.46
**Hipline (cm)**	113.60 ± 11.19	114.43 ± 12.03	115.71 ± 14.79
**Mean WHR (SD)**	0.75 ± 0.06	0.74 ± 0.07	0.76 ± 0.07
**Heartbeat (min)**	85.78 ± 16.64	82.31 ± 9.35	78.67 ± 15.29
**Mean PSQI (SD)**	7.58 ± 3.57	8.23 ± 3.34	7.08 ± 3.65
**Mean SWLS (SD)**	22.31 ± 6.52	21.01 ± 7.14	21.88 ± 6.55
**Mean SWB (SD)**	80.61 ± 9.49	77.77 ± 8.85	78.67 ± 10.49
**Mean GDS (SD)**	16.20 ± 4.34	17.14 ± 3.78	17.63 ± 3.81
**Mean GAI-20 (SD)**	13.62 ± 1.88	13.92 ± 1.76	13.96 ± 2.15
**Mean SMS (SD)**	5.83 ± 0.71	5.79 ± 0.76	5.92 ± 0.30

Yrs: years, Pittsburgh Sleep Quality Index: PSQI, Satisfaction with Life Scale: SWLS, Subjective Well-Being: SWB, Geriatric Depression Scale: GDS, Geriatric Anxiety Inventory-20: GAI-20, and Sport Motivation Scale: SMS. Waist-to-Hip Ratio: WHR; Sedentary leisure exercise group: SLE; Vital capacity, weight, and heart rate were measured as described in Methods.

**Table 3 brainsci-15-00918-t003:** Intergroup comparison of the effects of different types of exercise on the mental health of the elderly women.

Variable	Types of Exercises	Mean ± SD	Types of Exercises	Mean Deviation	SE	*p*	95% CI		*f*
							Lower Limit	Upper Limit	
Hight	Oriental	166.4 ± 5.0	General	0.023	0.472	0.812	−1.11	1.16	0.102
	General	166.55 ± 5.72	SLE	−1.771	0.934	0.175	−4.01	0.47	0.33
	SLE	167.63 ± 4.92	Oriental	0.263	0.789	0.253	−1.63	2.16	0.25
Weight	Oriental	64.16 ± 6.39	General	0.9209	0.5712	0.322	−0.45	2.292	0.135
	General	63.3 ± 6.33	SLE	−3.1992 *	1.13	0.014 *	−5.912	−0.487	0.748
	SLE	66.20 ± 6.72	Oriental	1.748	0.886	0.147	−0.38	3.87	0.371
BMI	Oriental	23.15 ± 1.90	General	0.1135	0.1723	0.489	−0.3	0.527	0.102
	General	23.00 ± 1.89	SLE	−0.3958	0.3409	0.738	−1.214	0.423	0.042
	SLE	23.50 ± 2.20	Oriental	2.2783	1.0722	0.102	−0.296	4.852	0.212
Vital Capacity	Oriental	2047.08 ± 214.04	General	250.329 *	35.783	0.001 **	164.43	336.23	0.62
	General	1916.11 ± 203.08	SLE	−337.291 *	70.791	0.022 *	−507.23	−167.35	0.52
	SLE	2021.13 ± 199.34	Oriental	0.2823	0.3235	0.550	−0.494	1.059	0.22
Waistline	Oriental	84.49 ± 8.96	General	0.061	0.778	0.915	−1.81	1.93	0.01
	General	84.38 ± 8.99	SLE	−0.911	1.539	0.432	−4.61	2.79	0.20
	SLE	86.21 ± 10.46	Oriental	86.962	67.172	0.588	−74.29	248.21	0.19
Hipline	Oriental	113.60 ± 11.19	General	−0.749	1.046	0.538	−3.26	1.76	0.07
	General	114.43 ± 12.03	SLE	−0.456	2.07	0.695	−5.43	4.51	0.10
	SLE	115.71 ± 14.79	Oriental	0.849	1.461	0.530	−2.66	4.36	0.18
WHR	Oriental	0.75 ± 0.06	General	0.00514	0.00548	0.194	−0.008	0.0183	0.17
	General	0.74 ± 0.07	SLE	−0.0061	0.01085	0.530	−0.0321	0.0199	0.14
	SLE	0.76 ± 0.07	Oriental	1.205	1.964	0.335	−3.51	5.92	0.17
Heartbeat	Oriental	85.78 ± 16.64	General	2.679	4.66	0.097	−8.51	13.87	0.19
	General	82.31 ± 9.35	SLE	2.488	9.219	0.166	−19.64	24.62	0.24
	SLE	78.67 ± 15.29	Oriental	−5.167	8.748	0.334	−26.17	15.83	0.14
PSQI	Oriental	7.58 ± 3.57	General	−0.734	0.341	0.096	−1.55	0.09	0.19
	General	8.23 ± 3.34	SLE	1.649 *	0.675	0.045 *	0.03	3.27	0.343
	SLE	7.08 ± 3.65	Oriental	−0.915	0.641	0.462	−2.45	0.62	0.14
SWLS	Oriental	22.31 ± 6.52	General	1.588	0.769	0.118	−0.26	3.43	0.19
	General	21.01 ± 7.14	SLE	−2.645	1.521	0.248	−6.3	1.01	0.12
	SLE	21.88 ± 6.55	Oriental	1.057	1.443	0.761	−2.41	4.52	0.07
SWB	Oriental	80.61 ± 9.49	General	3.168 *	1.19	0.024 *	0.31	6.02	0.341
	General	77.77 ± 8.85	SLE	−3.795	2.353	0.322	−9.44	1.85	0.10
	SLE	78.67 ± 10.49	Oriental	0.627	2.233	0.391	−4.73	5.99	0.20
GDS	Oriental	16.20 ± 4.34	General	−1.024 *	0.373	0.019 *	−1.92	−0.13	0.32
	General	17.14 ± 3.78	SLE	−0.403	0.738	0.571	−2.17	1.37	0.13
	SLE	17.63 ± 3.81	Oriental	1.427	0.7	0.126	−0.25	3.11	0.23
GAI-20	Oriental	13.62 ± 1.88	General	−0.207	0.17	0.674	−0.62	0.2	0.16
	General	13.92 ± 1.76	SLE	−0.195	0.337	0.933	−1	0.61	0.02
	SLE	13.96 ± 2.15	Oriental	0.402	0.32	0.629	−0.37	1.17	0.18
SMS	Oriental	5.83 ± 0.71	General	0.042	0.067	0.639	−0.12	0.2	0.05
	General	5.79 ± 0.76	SLE	−0.17	0.132	0.602	−0.49	0.15	0.185
	SLE	5.92 ± 0.30	Oriental	0.127	0.126	0.934	−0.17	0.43	0.131

Pittsburgh Sleep Quality Index: PSQI, Satisfaction with Life Scale: SWLS, Subjective Well-Being: SWB, Geriatric Depression Scale: GDS, Geriatric Anxiety Inventory-20: GAI-20, and Sport Motivation Scale: SMS. Body Mass Index: BMI. Waist-to-Hip Ratio: WHR. * *p* < 0.05, ** *p* < 0.01.

**Table 4 brainsci-15-00918-t004:** A comparison of sleep and mental health between different areas.

Variable	Levene’s Test		t	Sig. (2-Tailed)	Mean Difference	SE Difference	95% CI		*d*
	F	*p*					Lower Limit	Upper Limit	
PSQI	0.05	0.822	0.724	0.469	0.235	0.325	−0.403	0.873	0.02
SWLS	3.723	0.054	−0.819	0.413	−0.502	0.613	−1.706	0.702	0.04
SWB	6.727	0.01 *	0.384	0.701	0.327	0.851	−1.345	1.998	0.05
GDS	0.443	0.506	0.407	0.684	0.156	0.384	−0.598	0.911	0.08
GAI-20	1.038	0.309	−0.264	0.792	−0.046	0.175	−0.389	0.297	0.15
SMS	5.46	0.02 *	1.166	0.244	0.0798	0.0684	−0.0545	0.2141	0.01

Pittsburgh Sleep Quality Index: PSQI, Satisfaction with Life Scale: SWLS, Subjective Well-Being: SWB, Geriatric Depression Scale: GDS, Geriatric Anxiety Inventory-20: GAI-20, and Sport Motivation Scale: SMS. * *p* < 0.05.

## Data Availability

Data are available from the corresponding author upon reasonable request. The data are not publicly available due to privacy or ethical restrictions.

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
