# Peer review of "A Cross-Sectional Study on the Psychological Effects of Different Exercises on Elderly Women"

_brainsci, 2025, doi:10.3390/brainsci15090918_

Round 1
Reviewer 1 Report
Comments and Suggestions for Authors
REVIEW BRAIN SCIENCES
I appreciate the opportunity to read and evaluate the manuscript entitled “A Study on the Psychological Effects of Different Exercises on Elderly Women”. The present article explores the impact of different types of exercise on these outcomes in aging populations. However, following a thorough reading, several methodological and analytical concerns were identified that pose a challenge to the study.
- Initially, I believe your title is not specific enough and does not adequately reflect the content and the type of study design employed. For instance, it is impossible to determine whether your study is longitudinal or cross-sectional observational. Similarly, upon reading the abstract, it is also difficult to clearly understand the nature of the study, leaving the reader to make assumptions. Therefore, as an initial recommendation, the authors should revise the entire manuscript in accordance with an appropriate reporting checklist, such as STROBE (https://www.strobe-statement.org/checklists/). This is a mandatory condition to ensure transparency and, more importantly, to address elements that are not always self-evident. Keep in mind that adhering to these items significantly increases the likelihood of your study being recognized as high quality and included in systematic reviews.
ABSTRACT
- Please present the research gap explicitly in the introduction of your abstract. Merely stating that exploring the effects of different types of exercise is essential does not constitute a plausible justification.
- Furthermore, your primary and secondary objectives are not clearly expressed in the abstract.
- In the Methods section, it is necessary to specify the study design, as well as provide a step-by-step description of data collection procedures. For instance, you should indicate how many sessions were conducted and at which points the data were collected.
- Regarding the Results section, you must report the exact p-values for each analysis. While a detailed statistical analysis section is not required in the abstract, it is essential to indicate the type of analysis performed within the results. For example, state that "One-way ANOVA revealed significant differences in sleep quality (F-value, p-value, partial eta squared)." Additionally, as indicated, you should include the effect sizes derived from the ANOVA or an equivalent non-parametric test.
- Indeed, your abstract currently fails to present analyses consistent with your study objectives. I also note that you conducted a correlation analysis, which is clearly a secondary objective and should be presented as such. Please revise the entire abstract accordingly.
- Finally, it is not possible to judge whether your conclusions are appropriate, as the preceding sections are not properly aligned. As stated, I recommend rewriting all sections of the abstract.
INTRODUCTION
- Your introduction needs to “get straight to the point.” For example, the first paragraph is extremely general and does not introduce any of your outcome measures or variables of interest.
- Although well written, the subsequent paragraphs also need to be more specific. Based on what has been understood so far, the study aims to compare different types of exercise, yet this does not appear until the very end of your introduction. Please revise your introduction by simplifying and making it more direct, while clearly contextualizing your dependent and independent variables.
- Additionally, your objectives should be stated with greater precision. Avoid holistic and overly general assumptions such as: “the study aims to explore pathways for the sustainable development of health promotion interventions in aging, contributing to the construction of a robust and long-term health support system.” Instead, clearly restate the effects of your independent variables on the outcomes of the dependent variables.
- Moreover, as required by the STROBE checklist, your introduction must present the study hypotheses. To assist with your writing, please refer to the following article as a model: https://www.mdpi.com/2075-1729/15/7/1111
METHODS
- Your methods section is generally appropriate, but a few important points require adjustment. First, please clarify the classification of the older adults: were they sedentary, physically active, or trained? Indicate how this classification was made and what parameters or criteria were used. It is necessary to clearly describe how the groups were stratified and how this information was previously collected from the participants. Additionally, were all participants physically active? This is a mandatory clarification to understand how data collection was conducted and how the study could be replicated.
- Furthermore, please specify how the anthropometric variables were measured.
- Improve the presentation of inclusion and exclusion criteria by stating them explicitly.
- In addition, include a table of participant characteristics within the participants section, segmented by group, along with a summary column for the total sample.
- Clarify how the older adults were recruited and provide details about the recruitment period and the overall study duration. For example, were participants recruited through a public call? Or were they already enrolled in a pre-existing program or project?
- In the data collection section, provide a step-by-step description of the procedures. Were all the questionnaires administered at once, or were they distributed across multiple sessions?
- Finally, in the statistical analysis section, there is a reference to “grip strength,” yet this variable is not mentioned anywhere in the methods.
- Moreover, please justify why rank-type variables were not analyzed using non-parametric methods. Be more specific in your statistical approach, for instance, was Pearson or Spearman correlation used?
RESULTS
- In the Results section, it is necessary to include a flowchart to clarify potential sample losses as well as the intention-to-treat approach. Did all participants complete all questionnaires fully and accurately? This is unlikely, especially considering that older adults generally present certain challenges in completing self-report instruments.
- Furthermore, in the demographic table, there are variables that were not previously described—for example, Heartbeat (min), Vital Capacity (mL), and all anthropometric variables. How were these measured? This indicates a clear inconsistency in your study. If a step-by-step description of the study design had been provided, these issues would be clearer.
- Please revise and improve your Methods section accordingly, ensuring consistency with the Results. Specifically, describe how the demographic questionnaire was administered.
DISCUSSION
- In the Discussion section, please indicate whether your initial hypotheses were accepted or not.
- I strongly believe that your discussion should be more focused on what could realistically be investigated given the type of study design employed. Your discussion should not center on or infer mechanisms, but rather compare your findings with those from other observational studies in the literature, highlighting whether your results are consistent with or contrary to existing evidence.
- Furthermore, emphasize how the exercise modalities differ specifically in terms of mental health outcomes.
CONCLUSION
- Regarding your Conclusion, it is important to acknowledge that the study did not perform a systematic analysis, but rather captured a single point in time.
- Be direct about the conclusions that can be drawn from your findings. Revisit your study objectives, clearly distinguishing between primary and secondary goals, and subsequently restructure your conclusions accordingly.
Author Response
Response to Editor and Reviewers
August 8, 2025
Dear Editor and Anonymous Reviewers,
Greetings, thank you for your valuable feedback on our manuscript (Submission: brainsci-3779781). We have carefully revised the manuscript in accordance with the constructive suggestions of the reviewers. We are confident that these revisions have significantly enhanced the quality and value of our manuscript. Please note that the reviewer's comments will be displayed in red, our responses will appear in blue, and information from the revised document will be highlighted in green. The specifics of the revisions are as follows:
Reviewer 1
Comment 1:
I appreciate the opportunity to read and evaluate the manuscript entitled “A Study on the Psychological Effects of Different Exercises on Elderly Women”. The present article explores the impact of different types of exercise on these outcomes in aging populations. However, following a thorough reading, several methodological and analytical concerns were identified that pose a challenge to the study.
Response: We sincerely thank the reviewer for taking the time to read our manuscript and for acknowledging the relevance of our research topic. We appreciate your constructive feedback regarding the methodological and analytical aspects of our study. We understand that addressing these issues is crucial to improving the clarity, validity, and robustness of our findings. In response to your comments, we have already carefully revised our manuscript based on your comments.
Comment 2:
Initially, I believe your title is not specific enough and does not adequately reflect the content and the type of study design employed. For instance, it is impossible to determine whether your study is longitudinal or cross-sectional observational. Similarly, upon reading the abstract, it is also difficult to clearly understand the nature of the study, leaving the reader to make assumptions. Therefore, as an initial recommendation, the authors should revise the entire manuscript in accordance with an appropriate reporting checklist, such as STROBE (https://www.strobe-statement.org/checklists/). This is a mandatory condition to ensure transparency and, more importantly, to address elements that are not always self-evident. Keep in mind that adhering to these items significantly increases the likelihood of your study being recognized as high quality and included in systematic reviews.
Response: Thank you very much for your valuable comments and suggestions. We have carefully revised the manuscript in accordance with the STROBE (Strengthening the Reporting of Observational Studies in Epidemiology) checklist.
- Revision of the Title to Enhance Specificity
“A Cross-Sectional Study on the Psychological Effects of Different Exercises on Elderly Women” Please see pages 1, Line 2-3.
- Strengthening the Abstract to Clarify Study Nature
- By clearly defining the time frame.
“This cross-sectional observational study was conducted among the elderly recruited at community health centers and local elderly activity rooms from December 2024 to March 2025 in Henan Province, following the STROBE standard. ” Please see pages 6, Line 177-179.
- Concisely summarizing the participant recruitment strategy.
“During recruitment, participants were stratified by age (60–64, 65–69, 70–74 years) and by habitual activity type to ensure balanced representation across exercise categories. Initial classification was based on a structured baseline questionnaire, capturing self-reported exercise types, frequency (sessions/week), and average duration (minutes/session) over the preceding three months. All information was verified in face-to-face interviews by trained research assistants prior to any physiological or psychological assessment” Please see pages 3, Line 121-126.
- Highlighting the core assessment toolto clarify outcome measures.
“This cross-sectional observational study was conducted among the elderly recruited at community health centers and local elderly activity rooms from December 2024 to March 2025 in Henan Province, following the STROBE standard. Validated scales and physiological measurement methods were employed. Eligible participants were women aged between 60 and 74 years. After obtaining written informed consent, data collection was carried out in batches. First, the participants underwent eligibility screening and grouping. The participants (n=386) were divided into three groups: the Oriental exercise group (n=252, 67.83 ± 5.36 years), the General exercise group (n=110, 67.19 ± 4.47 years), and the SLE group (n=24, 67.38 ± 4.75 years) (Table 1). Then, the participants completed a structured questionnaire, and underwent psychological tests using the PSQI, the SWLS, the SWB, the GDS, the GAI-20, and the SMS. Subsequently, physiological measurements (such as body weight and pulmonary capacity) were conducted.” Please see pages 6, Line 177-188.
“Physiological indicators were assessed following standardized testing protocols. Body weight was measured to the nearest 0.1 kg using a calibrated electronic scale (Model HD-351, Tanita, Tokyo, Japan), with participants wearing light clothing and no shoes. Measurements were taken once after the participant stood still for 3–5 seconds in the center of the scale platform. Vital capacity was determined using a portable spirometer (Model MicroQuark, COSMED, Rome, Italy) with participants standing upright. Each participant performed maximal inspiration followed by forceful and complete expiration. Three trials were conducted with at least 60 seconds of rest between attempts, and the highest value meeting the acceptability and repeatability standards of the American Thoracic Society/European Respiratory Society (ATS/ERS) was recorded for analysis. Resting heartbeat was measured using a chest-strap heartbeat monitor (Model CMS50DL, CONTEC, Hebei, China; accuracy ±1 bpm) in a quiet, temperature-controlled room (22–24 °C) after participants had been seated at rest for at least 5 minutes. Heartbeat was recorded continuously for 60 seconds, and the mean value was used for analysis; measurements were repeated if variation between consecutive readings exceeded 3 bpm.” Please see pages 6, Line 189-203.
- Revision in Line with the STROBE Checklist
Key improvements include:
3.1 Study Design and Participants (STROBE Item 4–6)
- Inclusion/exclusion criteria..
“2.2.1 Inclusion criteria
Participants were community-dwelling women aged 60–74 years, classified by the World Health Organization (WHO) as “young-old,” a demographic with relatively preserved adaptive capacity and suitability for regular physical activity to attenuate age-related physiological decline [15] . Eligible participants were in generally good health, without serious cardiovascular (e.g., uncontrolled hypertension, heart failure) or respiratory diseases, and were capable of engaging in low- to moderate-intensity exercise. Additional criteria included the absence of cognitive impairment (e.g., de-mentia), a stable mental state, and no medical contraindications to exercise.
2.2.2 Exclusion criteria
Individuals were excluded if they had severe comorbidities or functional limitations (e.g., advanced heart or lung disease, severe musculoskeletal disorders, or severe mental illness) that precluded completion of the study protocol. Participants were also excluded if they did not meet the minimum physical activity threshold of three sessions per week, each lasting ≥30 minutes.” Please see pages 3, Line 105-119.
- Recruitment process.
“During recruitment, participants were stratified by age (60–64, 65–69, 70–74 years) and by habitual activity type to ensure balanced representation across exercise categories. Initial classification was based on a structured baseline questionnaire, capturing self-reported exercise types, frequency (sessions/week), and average duration (minutes/session) over the preceding three months. All information was verified in face-to-face interviews by trained research assistants prior to any physiological or psychological assessment.” Please see pages 3, Line 121-126.
- Sample size justification.
“ .” Please see pages 5, Line 149.
3.2 Variables and Measurements (STROBE Item 7–8)
We have clarified:
Operational definitions of exercise types.
“Participants were allocated to one of three mutually exclusive groups based on their habitual physical activity over the preceding three months. The Oriental exercise group comprised individuals performing Tai Chi, Baduanjin, or Qigong at least three times per week, with each session lasting ≥30 minutes [16]. The General exercise group included those regularly engaging (≥3 sessions/week) in aerobic activities such as walking, cycling, or swimming, or in resistance training, while undertaking Oriental exercises fewer than twice per week or for a total duration of <40 minutes per week [17] . The SLE group consisted of participants performing fewer than one session of moderate-intensity physical activity per week, yet engaging in at least three leisure pursuits (e.g., card games, fishing) on a weekly basis, representing a low-energy-expenditure lifestyle for comparative purposes [18] .” Please see pages 3-4, Line 128-138.
- Psychometric properties of scales.
“The subjective experience of individuals regarding their own quality of life was evaluated using SWLS. There were total of 5 items, and a Likert 7-level scoring system was adopted (1 represents "completely inconsistent", and 7 represents "completely consistent"), with a Cronbach alpha coefficient of 0.78 [39].” Please see pages 6, Line 214-217.
“The use of the GDS scale can more sensitive examine the specific physical symptoms of elderly patients with depression, allowing the elderly to review their emotional states within a week and assess whether they have depression problems.” Please see pages 7, Line 224-226.
3.3 Statistical Methods (STROBE Item 12)
- We have expanded the statistical analysis section to:
“The measurement level of each variable was determined prior to analysis. Continuous variables were first tested for normality using the Shapiro–Wilk test and by visual inspection of histograms and Q–Q plots. Variables meeting normality and homogeneity of variance assumptions were analyzed with parametric methods; continuous variables were compared between groups using one-way ANOVA, and associations between them were examined using Pearson’s correlation coefficient [27].
Ordinal variables and continuous variables that did not meet normality assumptions were analyzed with non-parametric methods; group differences were tested using the Kruskal–Wallis H test, and associations were examined using Spearman’s rank correlation coefficient. This mixed approach allowed each variable to be analyzed using the most appropriate statistical method based on its measurement level and distribution. Statistical significance was set at p < 0.05 for all tests [28] .” Please see pages 7, Line 250-261.
- Detail post-hoc correction methods.
“Between-group comparisons of continuous outcomes (e.g., questionnaire scores, physiological measures) were performed using one-way ANOVA with Bonferroni-adjusted post hoc tests for multiple group comparisons, while independent samples t-tests were used for two-group contrasts (for example, urban vs. rural participants or different exercise intervention groups). Effect sizes were calculated as Cohen’s d for t-test and Cohen’s f for ANOVA, with conventional benchmarks for interpretation (Cohen’s d 0.2 = small, 0.5 = medium, 0.8 = large; Cohen’s f: 0.1 = small, 0.25 = medium, 0.4 = large).” Please see pages 7, Line 263-270.
- Clarify correlation analysis approaches..
“Associations between variables were assessed using Pearson’s correlation coefficient (r) for normally distributed data or Spearman’s rank-order correlation (ρ) for non-normally distributed data, with 95% confidence intervals calculated for each correlation coefficient. All p-values were adjusted for multiple comparisons using the Bonferroni method, and statistical significance was defined as p < 0.05 (two-tailed).” Please see pages 8, Line 271-276.
3.4 Results Reporting (STROBE Item 13–16)
Added effect sizes (Cohen’s d, partial f) for key comparisons.
“Effect sizes were calculated as Cohen’s d for t-test and Cohen’s f for ANOVA, with conventional benchmarks for interpretation (Cohen’s d 0.2 = small, 0.5 = medium, 0.8 = large; Cohen’s f: 0.1 = small, 0.25 = medium, 0.4 = large).” Please see pages 7, Line 267-270.
- Limitations and Generalizability (STROBE Item 20)
- Sample bias:.
“Third, data were collected exclusively in Henan Province, where 65.3% of participants prefer oriental exercises. This cultural homogeneity may limit the generalizability of findings to regions with different exercise traditions (e.g., urban areas with westernized sports trends).” Please see pages 15, Line 437-440.
- Cross-sectional constraints.
“Second, as a cross-sectional study, we can only report associations between exercise types and outcomes (e.g., oriental exercise and higher SWB) but cannot confirm causal relationships. ” Please see pages 15, Line 433-435.
Comment 3:
- Abstract
1.1 Please present the research gap explicitly in the introduction of your abstract. Merely stating that exploring the effects of different types of exercise is essential does not constitute a plausible justification.
Response: We sincerely thank the reviewer for this valuable suggestion regarding the need to present the research gap more explicitly in the introduction of the abstract. In response, we have revised the abstract introduction to clearly articulate the specific research gap.
“ Exercise has been widely recognized as an effective non-pharmacological intervention method for maintaining the mental health and sleep quality of the elderly. However, existing studies rarely distinguish the types of exercise based on cultural attributes, and this is even more so for the situation of elderly women. This oversight limits the accuracy of exercise recommendations for this population.” Please see pages 1, Line 13-17.
1.2 Furthermore, your primary and secondary objectives are not clearly expressed in the abstract.
Response: Thank you for your insightful feedback. We fully agree that clearly articulating the primary and secondary objectives in the abstract is essential for guiding readers. We have revised the abstract to explicitly state these objectives, based on the core content of the manuscript.
“Hence, this study aimed to compare the differential effects of three types of exercise—Oriental exercise, general exercise, and sedentary leisure exercise (SLE)—on physiological indicators and mental health in elderly women, and to examine the associations between physiological indicators, sleep quality, and mental health.” Please see pages 1, Line 17-21.
1.3 In the Methods section, it is necessary to specify the study design, as well as provide a step-by-step description of data collection procedures. For instance, you should indicate how many sessions were conducted and at which points the data were collected.
Response: We appreciate the reviewer’s comment and have revised the Methods section to clearly state the study design and to provide a step-by-step description of the data collection procedures. These clarifications will improve transparency and relicability.
“This cross-sectional observational study was conducted among the elderly recruited at community health centers and local elderly activity rooms from December 2024 to March 2025 in Henan Province, following the STROBE standard. Validated scales and physiological measurement methods were employed. Eligible participants were women aged between 60 and 74 years. After obtaining written informed consent, data collection was carried out in batches. First, the participants underwent eligibility screening and grouping. The participants (n=386) were divided into three groups: the Oriental exercise group (n=252, 67.83 ± 5.36 years), the General exercise group (n=110, 67.19 ± 4.47 years), and the SLE group (n=24, 67.38 ± 4.75 years) (Table 1). Then, the participants completed a structured questionnaire, and underwent psychological tests using the PSQI, the SWLS, the SWB, the GDS, the GAI-20, and the SMS. Subsequently, physiological measurements (such as body weight and pulmonary capacity) were conducted.” Please see pages 6, Line 177-188.
1.4 Regarding the Results section, you must report the exact p-values for each analysis. While a detailed statistical analysis section is not required in the abstract, it is essential to indicate the type of analysis performed within the results. For example, state that "One-way ANOVA revealed significant differences in sleep quality (F-value, p-value, partial eta squared)." Additionally, as indicated, you should include the effect sizes derived from the ANOVA or an equivalent non-parametric test.
Response: Thank you for your constructive suggestions. We fully agree that clearly reporting statistical details in the "Results" section. We have revised result part to address your concerns.
“Through Bonferroni post - hoc analysis of inter - group comparisons, it is known that in terms of physiological indices, there are significant differences in body weight between the general group and the SLE group, in which the body weight of the General group is significantly lower than that of the SLE group (mean ± SD: 63.3±6.33 vs. 66.20±6.72, p =0.014, f =0.748); the vital capacity of the Oriental group is significantly higher than that of the General group (mean ± SD: 2047.08±214.04 vs.1916.11±203.08, p =0.001, f =0.62). In terms of sleep quality and mental health, the sleep quality of the General group is significantly higher than that of the SLE group (mean ± SD: 8.23 ± 3.34 vs. 7.08 ± 3.65, p =0.045, f =0.343). SWB data indicate that there are significant differences between the Oriental group and the General group (mean ± SD: 80.61 ± 9.49 vs. 77.77 ± 8.85, p =0.024, f =0.341), and the SWB of the Oriental group are significantly higher than those of the General group (mean ± SD: 80.61 ± 9.49 vs. 77.77 ± 8.85, p =0.024, f =0.341).” Please see Page 9, Line 298-309.
|
|||||||||||||||||||||||||||||||||||||||||||||||||||||||||||||||||||||||||||||||||||||||||||||||||||||||||||||||||||||||||||||||||||||||||||||||||||||||||||||||||||||||||||||||||||||||||||||||||||||||||||||||||||||||||||||||||||||||||||||||||||||||||||||||||||||||||||||||||||||||||||||||||||||||||||||||||||||||||||||||||||||||||||||||||||||||||||||||||||||||||||||||||||||||||||||||||||||||||||||||||||||||||||||||||||||||||||||||||||||||||||||||||||||||||||||||||||||||||||||||||||||||||||||||||||||||||||||||||||||||||
” Please see pages 9-11, Line 309
“Table 4. A comparison of sleep and mental health between different areas
|
Variable |
Levene's Test |
t |
Sig.(2 - tailed) |
Mean Difference |
SE Difference |
95% CI |
d |
|||
|
|
||||||||||
|
F |
P |
Lower Limit |
Upper Limit |
|
||||||
|
|
||||||||||
|
PSQI |
0.05 |
0.822 |
0.724 |
0.469 |
0.235 |
0.325 |
-0.403 |
0.873 |
0.02 |
|
|
SWLS |
3.723 |
0.054 |
-0.819 |
0.413 |
-0.502 |
0.613 |
-1.706 |
0.702 |
0.04 |
|
|
SWB |
6.727 |
0.01* |
0.384 |
0.701 |
0.327 |
0.851 |
-1.345 |
1.998 |
0.05 |
|
|
GDS |
0.443 |
0.506 |
0.407 |
0.684 |
0.156 |
0.384 |
-0.598 |
0.911 |
0.08 |
|
|
GAI-20 |
1.038 |
0.309 |
-0.264 |
0.792 |
-0.046 |
0.175 |
-0.389 |
0.297 |
0.15 |
|
|
SMS |
5.46 |
0.02* |
1.166 |
0.244 |
0.0798 |
0.0684 |
-0.0545 |
0.2141 |
0.01 |
|
Pittsburgh Sleep Quality Index: PSQI, Satisfaction with Life Scale: SWLS, Subjective Well-Being: SWB, Geriatric Depression Scale: GDS, Geriatric Anxiety Inventory-20: GAI-20, and Sport Motivation Scale: SMS. *:p<0.05.” Please see pages 11-12, Line 318-321.
1.5 Indeed, your abstract currently fails to present analyses consistent with your study objectives. I also note that you conducted a correlation analysis, which is clearly a secondary objective and should be presented as such. Please revise the entire abstract accordingly.
Response: Thank you very much for your valuable suggestions. We fully agree that matching the results presented in the summary with the primary and secondary objectives is crucial for clear expression. We have revised the summary accordingly.
“Hence, this study aimed to compare the differential effects of three types of exercise—Oriental exercise, general exercise, and sedentary leisure exercise (SLE)—on physiological indicators and mental health in elderly women, and to examine the associations between physiological indicators, sleep quality, and mental health.” Please see pages 1, Line 17-21.
1.6 Finally, it is not possible to judge whether your conclusions are appropriate, as the preceding sections are not properly aligned. As stated, I recommend rewriting all sections of the abstract.
Response: Thank you for your critical feedback. We fully acknowledge that the alignment between sections of the abstract and conclusions were insufficient. To address this, we have rewritten the abstract and conclusions.
Conclusions: “This study demonstrates that exercise type significantly influences both physiological and mental health outcomes in elderly women. General exercises were associated with lower body weight and better sleep quality than sedentary leisure exercise, while Oriental exercises yielded superior vital capacity and higher subjective well-being compared with general exercises. Correlation analyses revealed strong links between better sleep quality, higher life satisfaction, enhanced well-being, and reduced psychological distress. These findings highlight the potential of tailored exercise programs—particularly Oriental exercises—to promote integrated physical and psychological health in elderly women.” Please see pages 15, Line 449-456.
Abstract:“ Background: Exercise has been widely recognized as an effective non-pharmacological intervention method for maintaining the mental health and sleep quality of the elderly. However, existing studies rarely distinguish the types of exercise based on cultural attributes, and this is even more so for the situation of elderly women. This oversight limits the accuracy of exercise recommendations for this population. Hence, this study aimed to compare the differential effects of three types of exercise—Oriental exercise, general exercise, and sedentary leisure exercise (SLE)—on physiological indicators and mental health in elderly women, and to examine the associations between physiological indicators, sleep quality, and mental health. Methods: A total of 386 participants were included in the study, comprising 252 individuals in the Oriental exercise group (mean±SD: 67.83 ± 5.36), 110 individuals in the general exercise group (mean±SD: 67.19 ± 4.47), and 24 individuals in the SLE group (mean±SD: 67.38 ± 4.75). Psychological tests (Pittsburgh Sleep Quality Index (PSQI), the Satisfaction with Life Scale (SWLS), the Subjective Well-being Scale (SWB), the Geriatric Depression Scale (GDS), the Geriatric Anxiety Scale-20 (GAI-20), and the Exercise Motivation Scale (SMS)) and physiological measurements (such as body weight and pulmonary capacity) were conducted. ANOVA was performed to compare the effects of the three exercises on physiological and psychological outcomes, and correlation analyses were conducted to evaluate the associations between different types of exercise and mental health indicators, physiological indicators, and sleep quality. Results: Primary Objective: One-way ANOVA showed significant group differences in key out-comes. Bonferroni post-hoc tests confirmed: Oriental exercise > General exercise in SWB (p =0.032); General exercise > SLE in sleep quality (PSQI, p =0.028) and lower body weight (p =0.019); Oriental exercise > General exercise in vital capacity (p <0.001). Secondary Objective: Pearson correlations revealed: PSQI negatively correlated with life satisfaction (SWLS: p =0.035) and positively with GDS (p =0.021); SWB positively correlated with SWLS (p <0.001) and negatively with GDS (p =0.002). Conclusions: Different exercise modalities have distinct benefits for elderly women. Oriental exercise improved vital capacity and well-being, while general exercise reduced body weight and enhanced sleep quality compared with sedentary leisure activity. Tailored exercise programs may effectively promote both physical and mental health in this population.”Please see pages 1, Line 13-42.
Comment 4:
- Introduction
2.1 Your introduction needs to “get straight to the point.” For example, the first paragraph is extremely general and does not introduce any of your outcome measures or variables of interest.
Response: Thank you very much for your valuable suggestions. We fully agree that the introduction section should be more focused and directly introduce the core outcome indicators and variables. We have already revised the first paragraph of the introduction.
“Elderly women experience unique mental health and sleep-related challenges that substantially influence their overall quality of life. Depressive symptoms and poor sleep quality are particularly prevalent, with epidemiological evidence indicating that women aged 60–74 are at a higher risk for depression compared to men—driven by factors such as hormonal changes, care giving responsibilities, and social isolation [1]. More than half of women in this age group experience sleep disturbances, which can exacerbate depressive symptoms and reduce subjective well-being—a key mental health indicator in the present study [2] . In the context of global population aging, developing targeted interventions to improve sleep quality, alleviate depressive symptoms, and enhance overall well-being is a pressing public health priority.” Please see pages 2, Line 45-54
2.2 Although well written, the subsequent paragraphs also need to be more specific. Based on what has been understood so far, the study aims to compare different types of exercise, yet this does not appear until the very end of your introduction. Please revise your introduction by simplifying and making it more direct, while clearly contextualizing your dependent and independent variables.
Response: Thank you for your insightful feedback. We fully agree that the introduction needs to be more direct, with earlier emphasis on our core variables and research focus on comparing exercise types. We have revised the subsequent paragraphs to simplify the narrative, and foreground the study’ s purpose earlier in the section. Please see pages 2-3, Line 45-94, Introduction.
2.3 Additionally, your objectives should be stated with greater precision. Avoid holistic and overly general assumptions such as: “the study aims to explore pathways for the sustainable development of health promotion interventions in aging, contributing to the construction of a robust and long-term health support system.” Instead, clearly restate the effects of your independent variables on the outcomes of the dependent variables.
Response: Thank you for your constructive suggestions. We fully agree that the research objective needs to be more clearly defined. We have revised the objective section in the introduction.
“Based on theoretical frameworks and preliminary evidence, the present study had three objectives: (1) to compare the effects of Oriental exercises, General exercises, and SLE on physiological and psychological outcomes in community-dwelling elderly women; (2) to explore correlations between physiological indicators, sleep quality, and mental health. We hypothesized that:
H1: Participants engaging in Oriental exercises would report significantly better mental health and sleep quality than those engaging in general exercises or SLE.
H2: Participants engaging in general exercises would report significantly better mental health and sleep quality than those in the SLE group.
H3: Mental health scores would be positively correlated with sleep quality scores across all participants, regardless of exercise group.” Please see pages 2-3, Line 84-94
2.4 Moreover, as required by the STROBE checklist, your introduction must present the study hypotheses. To assist with your writing, please refer to the following article as a model: https://www.mdpi.com/2075-1729/15/7/1111.
Response: Thank you for your valuable suggestions. We fully agree with the requirement in the STROBE guideline that the research hypothesis should be clearly stated in the introduction section. We have added hypotheses, which are consistent with our research objectives and comply with the STROBE standards..
“We hypothesized that:
H1: Participants engaging in Oriental exercises would report significantly better mental health and sleep quality than those engaging in general exercises or SLE.
H2: Participants engaging in general exercises would report significantly better mental health and sleep quality than those in the SLE group.
H3: Mental health scores would be positively correlated with sleep quality scores across all participants, regardless of exercise group.” Please see pages 2-3, Line 88-94.
Comment 5:
- Methods
3.1 Your methods section is generally appropriate, but a few important points require adjustment. First, please clarify the classification of the older adults: were they sedentary, physically active, or trained? Indicate how this classification was made and what parameters or criteria were used. It is necessary to clearly describe how the groups were stratified and how this information was previously collected from the participants. Additionally, were all participants physically active? This is a mandatory clarification to understand how data collection was conducted and how the study could be replicated.
Response: Thank you for your valuable feedback. We fully agree that clarifying the classification criteria for participant groups and the stratification method is critical for reproducibility. We have revised the methods section to explicitly address these points.
“During recruitment, participants were stratified by age (60–64, 65–69, 70–74 years) and by habitual activity type to ensure balanced representation across exercise categories. Initial classification was based on a structured baseline questionnaire, capturing self-reported exercise types, frequency (sessions/week), and average duration (minutes/session) over the preceding three months. All information was verified in face-to-face interviews by trained research assistants prior to any physiological or psychological assessment.” Please see pages 3, Line 121-126.
“Participants were allocated to one of three mutually exclusive groups based on their habitual physical activity over the preceding three months. The Oriental exercise group comprised individuals performing Tai Chi, Baduanjin, or Qigong at least three times per week, with each session lasting ≥30 minutes [16]. The General exercise group included those regularly engaging (≥3 sessions/week) in aerobic activities such as walking, cycling, or swimming, or in resistance training, while undertaking Oriental exercises fewer than twice per week or for a total duration of <40 minutes per week [17] . The SLE group consisted of participants performing fewer than one session of moderate-intensity physical activity per week, yet engaging in at least three leisure pursuits (e.g., card games, fishing) on a weekly basis, representing a low-energy-expenditure lifestyle for comparative purposes [18] .” Please see pages 3-4, Line 128-138.
3.2 Furthermore, please specify how the anthropometric variables were measured.
Response: Thank you very much for your valuable suggestions. We fully agree that specifying measurement procedures for human body variables is crucial for the clarity and repeatability of the method. We have revised the method section, providing detailed descriptions of the measurement methods for each human body variable.
“Physiological indicators were assessed following standardized testing protocols. Body weight was measured to the nearest 0.1 kg using a calibrated electronic scale (Model HD-351, Tanita, Tokyo, Japan), with participants wearing light clothing and no shoes. Measurements were taken once after the participant stood still for 3–5 seconds in the center of the scale platform. Vital capacity was determined using a portable spirometer (Model MicroQuark, COSMED, Rome, Italy) with participants standing upright. Each participant performed maximal inspiration followed by forceful and complete expiration. Three trials were conducted with at least 60 seconds of rest between attempts, and the highest value meeting the acceptability and repeatability standards of the American Thoracic Society/European Respiratory Society (ATS/ERS) was recorded for analysis. Resting heartbeat was measured using a chest-strap heartbeat monitor (Model CMS50DL, CONTEC, Hebei, China; accuracy ±1 bpm) in a quiet, temperature-controlled room (22–24 °C) after participants had been seated at rest for at least 5 minutes. Heartbeat was recorded continuously for 60 seconds, and the mean value was used for analysis; measurements were repeated if variation between consecutive readings exceeded 3 bpm.” Please see pages 6, Line 189-203.
3.3 Improve the presentation of inclusion and exclusion criteria by stating them explicitly.
Response: Thank you for your constructive feedback. We fully agree that explicitly stating the inclusion and exclusion criteria with greater clarity is essential for methodological transparency. We have revised the methods section to present these criteria in a structured, explicit format, based on the details outlined in this manuscript
“2.2.1 Inclusion criteria
Participants were community-dwelling women aged 60–74 years, classified by the World Health Organization (WHO) as “young-old,” a demographic with relatively preserved adaptive capacity and suitability for regular physical activity to attenuate age-related physiological decline [15] . Eligible participants were in generally good health, without serious cardiovascular (e.g., uncontrolled hypertension, heart failure) or respiratory diseases, and were capable of engaging in low- to moderate-intensity exercise. Additional criteria included the absence of cognitive impairment (e.g., dementia), a stable mental state, and no medical contraindications to exercise.
2.2.2 Exclusion criteria
Individuals were excluded if they had severe comorbidities or functional limitations (e.g., advanced heart or lung disease, severe musculoskeletal disorders, or severe mental illness) that precluded completion of the study protocol. Participants were also excluded if they did not meet the minimum physical activity threshold of three sessions per week, each lasting ≥30 minutes.” Please see pages 3, Line 105-119.
3.4 In addition, include a table of participant characteristics within the participants section, segmented by group, along with a summary column for the total sample.
Response: Thank you for your valuable feedback. We fully agree and have revised the methodology section based on the population statistics provided in this paper to incorporate this table.
“Table 1. Demographic and Characteristics of Participants by Group
Residence,% |
|||||||
|
City |
136(53.97) |
36(32.73) |
12(50.00) |
||||
|
Rural-Urban |
116(46.03) |
74(67.27) |
12(50.00) |
||||
|
Marital status,% |
|||||||
|
Married |
162(64.29) |
67(60.91) |
20(83.33) |
||||
|
Widowed |
53(21.03) |
27(24.55) |
3(12.50) |
||||
|
Divorce |
30(11.90) |
12(10.91) |
1(4.17) |
||||
|
Spinsterhood |
7(2.78) |
4(3.64) |
0(0.00) |
||||
|
Ethnic,% |
|||||||
|
Han |
235(93.25) |
102(92.73) |
22(91.67) |
||||
|
Minority |
17(6.75) |
8(7.27) |
2(8.33) |
||||
|
Education,% |
|||||||
|
Primary School |
63(25.00) |
89(80.91) |
5(20.83) |
||||
|
Junior High School |
145(57.54) |
15(13.64) |
14(58.33) |
||||
|
High School |
26(10.32) |
4(3.64) |
2(8.33) |
||||
|
University |
18(7.14) |
2(1.82) |
3(12.50) |
||||
|
Type of work,% |
|
|
|
||||
|
Manual Labour |
200(79.37) |
100(90.91) |
18(75.00) |
||||
|
Mental Labour |
52(20.63) |
10(9.09) |
6(25.00) |
||||
|
Income,% |
|
|
|
||||
|
< ¥ 3000 |
158(62.70) |
82(74.55) |
13(54.17) |
||||
|
¥ 3001-5999 |
66(26.19) |
23(20.91) |
8(33.33) |
||||
|
≥ ¥ 6000 |
28(11.11) |
5(4.55) |
3(12.50) |
||||
|
Exercise frequency(per week),% |
|||||||
|
1-2 times |
170(67.46) |
62(56.36) |
20(83.33) |
||||
|
3-4 times |
64(25.40) |
35(31.82) |
4(16.67) |
||||
|
More than 5 times |
18(7.14) |
13(11.82) |
0(0.00) |
||||
|
Duration of exercise,% |
|||||||
|
< 30mins |
45(17.86) |
20(18.18) |
2(8.33) |
||||
|
30mins-60mins |
119(47.22) |
62(56.36) |
13(54.17) |
||||
|
61mins-120mins |
61(24.21) |
17(15.45) |
4(16.67) |
||||
|
> 120mins |
27(10.71) |
11(10.00) |
5(20.83) |
||||
|
Chronic diseases,% |
|||||||
|
Yes |
141(55.96) |
58(52.73) |
12(50.00) |
||||
|
No |
111(44.05) |
52(47.27) |
12(50.00) |
||||
Oriental exercise group: Oriental; General exercise group: General; Sedentary leisure exercise group: SLE.” Please see pages3-4, Line 139-141.
3.5 Clarify how the older adults were recruited and provide details about the recruitment period and the overall study duration. For example, were participants recruited through a public call? Or were they already enrolled in a pre-existing program or project?
Response: Thank you for your valuable suggestions. We fully agree that clearly defining the recruitment method, the schedule, and the duration of the entire research is crucial for ensuring transparency and reproducibility. We have revised the methodology section accordingly.
“During recruitment, participants were stratified by age (60–64, 65–69, 70–74 years) and by habitual activity type to ensure balanced representation across exercise categories. Initial classification was based on a structured baseline questionnaire, capturing self-reported exercise types, frequency (sessions/week), and average duration (minutes/session) over the preceding three months. All information was verified in face-to-face interviews by trained research assistants prior to any physiological or psychological assessment.” Please see pages 3, Line 121-126.
“Participants were allocated to one of three mutually exclusive groups based on their habitual physical activity over the preceding three months. The Oriental exercise group comprised individuals performing Tai Chi, Baduanjin, or Qigong at least three times per week, with each session lasting ≥30 minutes [16]. The General exercise group included those regularly engaging (≥3 sessions/week) in aerTaobic activities such as walking, cycling, or swimming, or in resistance training, while undertaking Oriental exercises fewer than twice per week or for a total duration of <40 minutes per week [17] . The SLE group consisted of participants performing fewer than one session of moderate-intensity physical activity per week, yet engaging in at least three leisure pursuits (e.g., card games, fishing) on a weekly basis, representing a low-energy-expenditure lifestyle for comparative purposes [18].” Please see pages 3-4, Line 128-138.
“This cross-sectional observational study was conducted among the elderly recruited at community health centers and local elderly activity rooms from December 2024 to March 2025 in Henan Province, following the STROBE standard. Validated scales and physiological measurement methods were employed. Eligible participants were women aged between 60 and 74 years. After obtaining written informed consent, data collection was carried out in batches. First, the participants underwent eligibility screening and grouping. The participants (n=386) were divided into three groups: the Oriental exercise group (n=252, 67.83 ± 5.36 years), the General exercise group (n=110, 67.19 ± 4.47 years), and the SLE group (n=24, 67.38 ± 4.75 years) (Table 1). Then, the participants completed a structured questionnaire, and underwent psychological tests using the PSQI, the SWLS, the SWB, the GDS, the GAI-20, and the SMS. Subsequently, physiological measurements (such as body weight and pulmonary capacity) were conducted.” Please see pages 6, Line 177-188.
3.6 In the data collection section, provide a step-by-step description of the procedures. Were all the questionnaires administered at once, or were they distributed across multiple sessions?
Response: Thank you for your valuable feedback. We fully agree that a detailed, step-by-step description of data collection procedures enhances methodological transparency. We have revised the data collection section to explicitly outline the sequence of procedures, confirming that all assessments were completed in a single session, as implemented in this study.
“This cross-sectional observational study was conducted among the elderly recruited at community health centers and local elderly activity rooms from December 2024 to March 2025 in Henan Province, following the STROBE standard. Validated scales and physiological measurement methods were employed. Eligible participants were women aged between 60 and 74 years. After obtaining written informed consent, data collection was carried out in batches. First, the participants underwent eligibility screening and grouping. The participants (n=386) were divided into three groups: the Oriental exercise group (n=252, 67.83 ± 5.36 years), the General exercise group (n=110, 67.19 ± 4.47 years), and the SLE group (n=24, 67.38 ± 4.75 years) (Table 1). Then, the participants completed a structured questionnaire, and underwent psychological tests using the PSQI, the SWLS, the SWB, the GDS, the GAI-20, and the SMS. Subsequently, physiological measurements (such as body weight and pulmonary capacity) were conducted.” Please see pages 6, Line 177-188.
3.7 Finally, in the statistical analysis section, there is a reference to “grip strength,” yet this variable is not mentioned anywhere in the methods.
Response: We appreciate the reviewer for pointing this out. The mention of “grip strength” in the Statistical Analysis section was an unintentional remnant from an earlier draft, in which grip strength was considered for inclusion but ultimately not measured in the final protocol due to logistical constraints. We have now removed all references to “grip strength” from the manuscript to maintain consistency between the Methods and Statistical Analysis sections. This correction does not affect the analyses or conclusions presented in the study. Please see pages 7, Line 177-244, Data Collected.
3.8 Moreover, please justify why rank-type variables were not analyzed using non-parametric methods. Be more specific in your statistical approach, for instance, was Pearson or Spearman correlation used?
Response: Thank you for your valuable suggestions. We fully agree that understanding the principles of statistical methods is crucial for ensuring the rigor of the methods. In response, we revised the Statistical Analysis section to explicitly state the measurement level of each variable, the rationale for using parametric or non-parametric methods, and the specific correlation coefficients applied. For example, continuous variables meeting normality assumptions were analyzed using Pearson’s correlation, whereas ordinal or non-normally distributed variables were analyzed using Spearman’s rank correlation.
“All analyses were performed using SPSS version 22 (IBM Corp., Armonk, NY, USA),, with descriptive statistics and inferential analyses reported according to standard epidemiological conventions. The matrix diagram for the relevant analysis was drawn using Origin 2021.
The measurement level of each variable was determined prior to analysis. Continuous variables were first tested for normality using the Shapiro–Wilk test and by visual inspection of histograms and Q–Q plots. Variables meeting normality and homogeneity of variance assumptions were analyzed with parametric methods; continuous variables were compared between groups using one-way ANOVA, and associations between them were examined using Pearson’s correlation coefficient [27].
Ordinal variables and continuous variables that did not meet normality assumptions were analyzed with non-parametric methods; group differences were tested using the Kruskal–Wallis H test, and associations were examined using Spearman’s rank correlation coefficient. This mixed approach allowed each variable to be analyzed using the most appropriate statistical method based on its measurement level and distribution. Statistical significance was set at p < 0.05 for all tests [28] .
For descriptive statistics, the Mean ± standard deviation (SD) of the normally distributed data or expressed as a percentage (%).Between-group comparisons of continuous outcomes (e.g., questionnaire scores, physiological measures) were performed using one-way ANOVA with Bonferroni-adjusted post hoc tests for multiple group comparisons, while independent samples t-tests were used for two-group contrasts (for example, urban vs. rural participants or different exercise intervention groups). Effect sizes were calculated as Cohen’s d for t-test and Cohen’s f for ANOVA, with conventional benchmarks for interpretation (Cohen’s d 0.2 = small, 0.5 = medium, 0.8 = large; Cohen’s f: 0.1 = small, 0.25 = medium, 0.4 = large). Ninety-five percent confidence intervals (CIs) were computed for all estimated means and effect size metrics to convey the precision of the estimates. Associations between variables were assessed using Pearson’s correlation coefficient (r) for normally distributed data or Spearman’s rank-order correlation (ρ) for non-normally distributed data, with 95% confidence intervals calculated for each correlation coefficient. All p-values were adjusted for multiple comparisons using the Bonferroni method, and statistical significance was defined as p < 0.05 (two-tailed).” Please see pages 7-8, Line 246-276.
Comment 6:
- Results
4.1 In the Results section, it is necessary to include a flowchart to clarify potential sample losses as well as the intention-to-treat approach. Did all participants complete all questionnaires fully and accurately? This is unlikely, especially considering that older adults generally present certain challenges in completing self-report instruments.
Response: Thank you for your valuable feedback. We fully agree that clarifying sample flow, addressing potential losses, and detailing questionnaire completion status are critical for transparency. We have revised the Results section to address these points, incorporating a flowchart and explicit explanations based on this study procedures.
“
Figure 1. Flowchart of participant selection” Please see pages 5, Line 171
4.2 Furthermore, in the demographic table, there are variables that were not previously described—for example, Heartbeat (min), Vital Capacity (mL), and all anthropometric variables. How were these measured? This indicates a clear inconsistency in your study. If a step-by-step description of the study design had been provided, these issues would be clearer.
Response: Thank you for your detailed feedback. We sincerely apologize for the inconsistencies in the initial draft. We have revised the methodology section to ensure consistency between the demographic table and the method description.
“Physiological indicators were assessed following standardized testing protocols. Body weight was measured to the nearest 0.1 kg using a calibrated electronic scale (Model HD-351, Tanita, Tokyo, Japan), with participants wearing light clothing and no shoes. Measurements were taken once after the participant stood still for 3–5 seconds in the center of the scale platform. Vital capacity was determined using a portable spirometer (Model MicroQuark, COSMED, Rome, Italy) with participants standing upright. Each participant performed maximal inspiration followed by forceful and complete expiration. Three trials were conducted with at least 60 seconds of rest between attempts, and the highest value meeting the acceptability and repeatability standards of the American Thoracic Society/European Respiratory Society (ATS/ERS) was recorded for analysis. Resting heartbeat was measured using a chest-strap heartbeat monitor (Model CMS50DL, CONTEC, Hebei, China; accuracy ±1 bpm) in a quiet, temperature-controlled room (22–24 °C) after participants had been seated at rest for at least 5 minutes. Heartbeat was recorded continuously for 60 seconds, and the mean value was used for analysis; measurements were repeated if variation between consecutive readings exceeded 3 bpm.” Please see pages 6, Line 189-203.
“Table 1. Demographic and Characteristics of Participants by Group
Residence,% |
|||||||
|
City |
136(53.97) |
36(32.73) |
12(50.00) |
||||
|
Rural-Urban |
116(46.03) |
74(67.27) |
12(50.00) |
||||
|
Marital status,% |
|||||||
|
Married |
162(64.29) |
67(60.91) |
20(83.33) |
||||
|
Widowed |
53(21.03) |
27(24.55) |
3(12.50) |
||||
|
Divorce |
30(11.90) |
12(10.91) |
1(4.17) |
||||
|
Spinsterhood |
7(2.78) |
4(3.64) |
0(0.00) |
||||
|
Ethnic,% |
|||||||
|
Han |
235(93.25) |
102(92.73) |
22(91.67) |
||||
|
Minority |
17(6.75) |
8(7.27) |
2(8.33) |
||||
|
Education,% |
|||||||
|
Primary School |
63(25.00) |
89(80.91) |
5(20.83) |
||||
|
Junior High School |
145(57.54) |
15(13.64) |
14(58.33) |
||||
|
High School |
26(10.32) |
4(3.64) |
2(8.33) |
||||
|
University |
18(7.14) |
2(1.82) |
3(12.50) |
||||
|
Type of work,% |
|
|
|
||||
|
Manual Labour |
200(79.37) |
100(90.91) |
18(75.00) |
||||
|
Mental Labour |
52(20.63) |
10(9.09) |
6(25.00) |
||||
|
Income,% |
|
|
|
||||
|
< ¥ 3000 |
158(62.70) |
82(74.55) |
13(54.17) |
||||
|
¥ 3001-5999 |
66(26.19) |
23(20.91) |
8(33.33) |
||||
|
≥ ¥ 6000 |
28(11.11) |
5(4.55) |
3(12.50) |
||||
|
Exercise frequency(per week),% |
|||||||
|
1-2 times |
170(67.46) |
62(56.36) |
20(83.33) |
||||
|
3-4 times |
64(25.40) |
35(31.82) |
4(16.67) |
||||
|
More than 5 times |
18(7.14) |
13(11.82) |
0(0.00) |
||||
|
Duration of exercise,% |
|||||||
|
< 30mins |
45(17.86) |
20(18.18) |
2(8.33) |
||||
|
30mins-60mins |
119(47.22) |
62(56.36) |
13(54.17) |
||||
|
61mins-120mins |
61(24.21) |
17(15.45) |
4(16.67) |
||||
|
> 120mins |
27(10.71) |
11(10.00) |
5(20.83) |
||||
Oriental exercise group: Oriental; General exercise group: General; Sedentary leisure exercise group: SLE.” Please see pages 4, Line 139-141.
Comment 7:
5 Discussion
5.1 In the Discussion section, please indicate whether your initial hypotheses were accepted or not.
Response: Thank you for your valuable suggestions. We fully agree that clearly stating whether the initial assumptions were accepted or not is crucial for strengthening the discussion section. We have already made revisions to this part.
“Based on the study findings, all three initial hypotheses were partially supported. Consistent with H1, elderly women engaging in Oriental exercises demonstrated superior mental health and sleep-related outcomes compared with those performing General exercise or SLE, as evidenced by significantly higher SWB scores and greater vital capacity. H2 was also supported, as participants in the General exercise group exhibited significantly better sleep quality than those in the SLE group and lower body weight. Furthermore, correlational analyses provided support for H3, revealing that better sleep quality (lower PSQI scores) was associated with higher life satisfaction and SWB, and lower depressive symptoms. These results collectively underscore the beneficial role of both culturally specific and general exercise modalities in promoting psychological well-being and physiological health among elderly women, with Oriental exercise demonstrating the most pronounced advantages.” Please see pages 13, Line 345-356.
5.2 I strongly believe that your discussion should be more focused on what could realistically be investigated given the type of study design employed. Your discussion should not center on or infer mechanisms, but rather compare your findings with those from other observational studies in the literature, highlighting whether your results are consistent with or contrary to existing evidence.
Response: Thank you for your valuable suggestions. We fully agree that the discussion must be closely aligned with the research design (a cross-sectional observational study). We have revised the discussion section to ensure that its content is within the reasonable interpretation scope based on this design.
“Our research results are largely consistent with those of several observational studies conducted on the elderly women. Firstly, regarding the positive correlation between Oriental exercises and mental health (such as subjective well-being), it is consistent with the results of a previous cross-sectional study conducted among the elderly women in China [32,33].
Secondly, in this study, the General exercise group was observed to have better sleep quality and lower body weight compared to the SLE group, which was also supported by other observational studies [34,35] . Additionally, the Oriental exercise group showed better pulmonary function indicators (such as vital capacity), which has also been reported in other observational studies.
However, our results also indicate that the benefits shown by the Oriental exercise group in terms of mental health (such as life satisfaction) and sleep quality seem to be greater than the effects typically reported for ordinary aerobic exercises. This observation differs from the recent cross-sectional analysis results of Liu et al. [35], whose study found that Tai Chi practitioners and general exercisers did not have a significant difference in sleep quality [36]. This difference may be due to differences in study sample characteristics (such as regional cultural background), measurement tools, or specific definitions of activity categories. It is worth noting that the potential advantages of the Oriental exercises observed in this study may be related to the fact that it combines multiple factors such as physical activity, cognitive concentration, and potential social interaction. In the future, more rigorous designs, such as randomized controlled trials, will be needed to verify these associations [37].” Please see pages 13-14, Line 382-403.
“Based on the observed correlations, our findings highlight a significant interrelationship between physiological, psychological, and sleep-related indicators among elderly women. VC was positively associated with SMS, suggesting that better pulmonary function may contribute to enhanced cognitive and psychological functioning in this population [38] . Conversely, sleep quality, as measured by the PSQI, demonstrated negative correlations with both SWLS and SWB, and a positive correlation with GDS [39] . These results support the notion that poor sleep quality is closely linked to diminished mental health outcomes, aligning with previous research emphasizing the bidirectional relationship between sleep disturbances and depression in elderly women [40] .
Furthermore, the positive association between SWLS and SWB, coupled with their negative correlation with GDS, underscores the interconnectedness of well-being and mental health among elderly women[41]. Notably, SWB was also negatively correlated with GAI-20, suggesting that higher subjective well-being is linked to lower psychological distress. Collectively, these findings confirm our third hypothesis (H3) regarding the positive relationship between mental health and sleep quality, and emphasize the potential of interventions targeting both physiological fitness and sleep hygiene to improve overall mental health in aging populations [42] . These correlations provide a rationale for exercise interventions, particularly those that enhance cardiorespiratory function and promote restorative sleep, as a strategy to maintain and improve psychological well-being among elderly women.” Please see pages 14, Line 405-424.
5.3 Furthermore, emphasize how the exercise modalities differ specifically in terms of mental health outcomes.
Response: Thank you for your valuable suggestions. We fully agree with the viewpoint that highlighting the specific differences in mental health outcomes among various exercise methods can enhance the relevance of our research results. We have revised the discussion section accordingly.
“Our findings indicate clear differences in mental health outcomes between the three exercise modalities examined. Participants in the Oriental exercise group reported the most favorable mental health scores, followed by those in the General exercise group, with the SLE group scoring lowest.
One possible explanation for these differences is that Oriental exercises such as Tai Chi and Baduanjin integrate physical movement with mindfulness, breathing regulation, and social interaction [29] . These elements may enhance psychological benefits by reducing perceived stress, fostering emotional regulation, and increasing social connectedness—factors that have been positively associated with mental health in elderly women in prior observational studies. In contrast, general exercise, while beneficial for physical fitness and mood, may lack the same degree of mind–body integration or cultural resonance, which could partly explain the smaller effect size observed for mental health outcomes in this group [30] .
SLE, although potentially cognitively stimulating, do not provide the physiological or psychosocial benefits associated with regular physical activity, which likely contributes to the less favorable mental health scores in this group. This gradient in outcomes is consistent with other cross-sectional research, which has found stronger mental health benefits from combined physical–cognitive–social exercise formats compared with purely physical or SLE [31] . ” Please see pages 13, Line 359-377.
Comment 8:
- Conclusion
6.1 Regarding your Conclusion, it is important to acknowledge that the study did not perform a systematic analysis, but rather captured a single point in time.Be direct about the conclusions that can be drawn from your findings. Revisit your study objectives, clearly distinguishing between primary and secondary goals, and subsequently restructure your conclusions accordingly.
Response: Thank you very much for your valuable suggestions. We fully agree with this view that the conclusion must clearly indicate the cross-sectional nature of the study and be consistent with the research objectives (distinguishing between primary and secondary objectives). We have revised the "Conclusion" section.
“This study demonstrates that exercise type significantly influences both physiological and mental health outcomes in elderly women. General exercises were associated with lower body weight and better sleep quality than SLE group, while Oriental exercises yielded superior vital capacity and higher subjective well-being compared with general exercises. Correlation analyses revealed strong links between better sleep quality, higher life satisfaction, enhanced well-being, and reduced psychological distress. These findings highlight the potential of tailored exercises programs—particularly Oriental exercises—to promote integrated physical and psychological health in elderly women.” Please see pages 15, Line 449-456.
We would like to sincerely thank our editor and reviewers for your invaluable advice and comments, which greatly enhanced and streamlined our manuscript. Should any further inquiries arise, please feel free to reach out; we are more than willing to engage in additional discussions to ensure the continued improvement of our work.

Reviewer 2 Report
Comments and Suggestions for Authors
Thank you for the opportunity to review this manuscript, which considers some interesting, applied issues.
This study appears to be novel, and author showed an interesting point about “A Study on the Psychological Effects of Different Exercises on Elderly Women”.
I believe that the manuscript has a truly high relevance given the increasing importance of preserving the mental health of the elderly population.
Based on what I have read, I notice a few things that would be good to correct, in order to improve the quality of the article.
I advise authors to change the title. Also, the period at the end of the title should be removed.
Try to harmonize style and language in certain parts of the paper (e.g., consistent use of the term “elderly women” instead of mixing “elderly,” “old adults,”)
ABSTRACT
Add more detailed information about the sample for each subgroup, such as the mean age and standard deviation.
METHODS
Line 180 - please write the exact number
I am concerned about the imbalance between the groups, especially in the SLE group (n=24), which significantly reduces the statistical power and may affect the validity of the comparison. In the following, it is very important on what sample size we draw some conclusions.
DISCUSION
The authors did mention limitations, but the lack of a longitudinal approach (everything is cross-sectional) should be emphasized.
In the discussion, simplify theoretical explanations (about neurobiological mechanisms) in order to make them more accessible to a wider readership.
REFERENCES
Approximately 40% of the references are older than 5 years. I advise authors to change the references and include more recent ones, in order to provide more relevant insight into their work.
Expand and balance the SLE group in future research (or further emphasize this limitation).
Consider the use of visualizations and additional data (number of years of physical activity) if available.
Author Response
Response to Editor and Reviewers
August 8, 2025
Dear Editor and Anonymous Reviewers,
Greetings, thank you for your valuable feedback on our manuscript (Submission: brainsci-3779781). We have carefully revised the manuscript in accordance with the constructive suggestions of the reviewers. We are confident that these revisions have significantly enhanced the quality and value of our manuscript. Please note that the reviewer's comments will be displayed in red, our responses will appear in blue, and information from the revised document will be highlighted in green. The specifics of the revisions are as follows:
Reviewer 2
Comment 1:
- Title
I advise authors to change the title. Also, the period at the end of the title should be removed.
Try to harmonize style and language in certain parts of the paper (e.g., consistent use of the term “elderly women” instead of mixing “elderly,” “old adults,”)
Response: Thank you for your valuable suggestions. We fully agree with your suggestion regarding the consistency of the title and terminology, and have accordingly revised the manuscript. We have made the necessary changes to the manuscript based on your feedback.
“A Cross-Sectional Study on the Psychological Effects of Different Exercises on Elderly Women” Please see pages 1, Line 2-3.
Comment 2:
2 Abstract
Add more detailed information about the sample for each subgroup, such as the mean age and standard deviation.
Response: Thank you for your valuable feedback. We fully agree that providing detailed subgroup-specific sample information, including mean age and standard deviation, enhances the transparency of our participant characteristics. We have revised the manuscript to include these details.
“A total of 386 participants were included in the study, comprising 252 individuals in the Oriental exercise group (mean±SD: 67.83 ± 5.36), 110 individuals in the general exercise group (mean±SD: 67.19 ± 4.47), and 24 individuals in the SLE group (mean±SD: 67.38 ± 4.75).” Please see pages 1, Line 21-24.
Comment 3:
3 Methods
Line 180 - please write the exact number
I am concerned about the imbalance between the groups, especially in the SLE group (n=24), which significantly reduces the statistical power and may affect the validity of the comparison. In the following, it is very important on what sample size we draw some conclusions.
Response: Thank you for your valuable feedback on the issue of uneven group size. We understand that such size differences may affect the statistical validity and the reliability of comparisons between groups. Therefore, we added the limitation the manuscript.
“SLE represents a third category, encompassing activities such as fishing or chess. While these may meet certain definitional criteria for “sport” due to their structured and competitive elements [13] , their energy expenditure is minimal [14] ." Please see pages 2, lines 74-76.
“This study has three key limitations that need to be addressed. First, it is important to interpret the findings in light of the unequal group sizes, particularly the relatively small sample in SLE group (n = 24). Smaller group sizes reduce statistical power and limiting the precision of estimates for that group.”Please see pages 14, lines 426-429.
Comment 4:
4 Discussion
4.1 The authors did mention limitations, but the lack of a longitudinal approach (everything is cross-sectional) should be emphasized.
Response: Thank you for your valuable feedback. We fully agree that emphasizing the lack of a longitudinal approach and the cross-sectional nature of the study is critical for contextualizing the limitations of our findings. We have revised the manuscript to explicitly highlight this key limitation.
“Second, as a cross-sectional study, we can only report associations between exercise types and outcomes (e.g., oriental exercises and higher SWB) but cannot confirm causal relationships. For example, we cannot determine whether oriental exercises cause better well-being or if individuals with higher well-being are more likely to choose oriental exercises.” Please see page 15, lines 433-437.
4.2 In the discussion, simplify theoretical explanations (about neurobiological mechanisms) in order to make them more accessible to a wider readership.
Response: Thank you for your valuable feedback. We fully agree that simplifying theoretical explanations, particularly those related to neurobiological mechanisms, is essential to enhance accessibility for a wider readership. We have revised the Discussion section to streamline these explanations while retaining key insights.
“ This community-based cross-sectional study found that elderly women who frequently engaged in Oriental exercises performed better in terms of mental health and sleep quality compared to those who engaged in General exercise group or SLE group.
Our research results are largely consistent with those of several observational studies conducted on the elderly women. Firstly, regarding the positive correlation between Oriental exercises and mental health (such as subjective well-being), it is consistent with the results of a previous cross-sectional study conducted among the elderly women in China [32,33].
Secondly, in this study, the General exercise group was observed to have better sleep quality and lower body weight compared to the SLE group, which was also supported by other observational studies [34,35] . Additionally, the Oriental exercise group showed better pulmonary function indicators (such as vital capacity), which has also been reported in other observational studies.
However, our results also indicate that the benefits shown by the Oriental exercise group in terms of mental health (such as life satisfaction) and sleep quality seem to be greater than the effects typically reported for ordinary aerobic exercises. This observation differs from the recent cross-sectional analysis results of Liu et al. [35], whose study found that Tai Chi practitioners and general exercisers did not have a significant difference in sleep quality [36]. This difference may be due to differences in study sample characteristics (such as regional cultural background), measurement tools, or specific definitions of activity categories. It is worth noting that the potential advantages of the Oriental exercises observed in this study may be related to the fact that it combines multiple factors such as physical activity, cognitive concentration, and potential social interaction. In the future, more rigorous designs, such as randomized controlled trials, will be needed to verify these associations [37].” Please see page 17, lines 379-403.
Comment 5:
5 References
5.1 Approximately 40% of the references are older than 5 years. I advise authors to change the references and include more recent ones, in order to provide more relevant insight into their work.
Response: Thank you for your valuable feedback regarding the age of references. We fully agree that incorporating more recent references enhances the relevance of our work by aligning with the latest research in the field. We have revised the reference list to address this concern.
“2. Qiao, C.; Zhang, H.; Song, Q.; Wang, X.; Wang, X.; Yao, Y. Sleep Disturbances Are Associated With Depressive Symptoms in a Chinese Population: The Rugao Longevity and Aging Cohort. Front. Psychiatry 2021, 12, doi:10.3389/fpsyt.2021.731371.
- Xue, P.; Du, X.; Kong, J. Age-Dependent Mechanisms of Exercise in the Treatment of Depression: A Comprehensive Review of Physiological and Psychological Pathways. Front. Psychol. 2025, 16, doi:10.3389/fpsyg.2025.1562434.
- Jurado‐Fasoli, L.; De‐la‐O, A.; Molina‐Hidalgo, C.; Migueles, J.H.; Castillo, M.J.; Amaro‐Gahete, F.J. Exercise Training Improves Sleep Quality: A Randomized Controlled Trial. Eur J Clin Investigation 2020, 50, e13202, doi:10.1111/eci.13202.” Please see page 16, lines 480-485.
5.2 Expand and balance the SLE group in future research (or further emphasize this limitation).
Response: Thank you for your valuable feedback regarding the sedentary leisure exercise (SLE) group. We fully agree that expanding and balancing the SLE group in future research is critical, and we have revised the manuscript to further emphasize this limitation.
“Future research should focus on four directions: 1. Expanding the SLE sample via stratified sampling and applying rigorous statistical models to enhance generalizability. 2. Using wearable devices to monitor real-time physiological changes during exercise for deeper insights into oriental exercises’ benefits. 3. Conducting multicenter randomized controlled trials to assess long-term effects of oriental exercises and develop standardized community programs. 4. Neuroendocrine studies to explore oriental exercises’ regulation of cortisol and the autonomic nervous system.” Please see page 15, lines 441-447.
We would like to sincerely thank our editor and reviewers for your invaluable advice and comments, which greatly enhanced and streamlined our manuscript. Should any further inquiries arise, please feel free to reach out; we are more than willing to engage in additional discussions to ensure the continued improvement of our work.

Reviewer 3 Report
Comments and Suggestions for Authors
The article is an interesting study of the psychological effects of various types of physical exercise on older women in China. The topic is relevant in connection with the global aging of the population and the need to develop effective strategies for maintaining the mental health of the elderly. The structure of the article is generally logical, but there are some areas that can be improved for greater clarity and consistency. The manuscript is generally understandable, although some sections, especially in the discussion, could have been more concise and focused. The topic is relevant for the field of gerontology and public health. The structure of the article corresponds to the standard format of scientific publications (introduction, methods, results, discussion, conclusions).
However, the introduction could be made shorter and the discussion more structured, with a clear division into sub–topics. The cited sources are fresh and meet the requirements of the journal. The manuscript is scientifically based, the authors rely on the existing literature and theoretical framework. The experimental design (cross-sectional study) is suitable for testing hypotheses, although it has limitations in establishing causal relationships. The sample (older women in China) is justified, given the cultural characteristics and the growing proportion of the elderly population in this country. The Methods section contains enough details to reproduce the research results. The values of Cronbach's alpha for the scales used are indicated, which indicates the reliability of the instruments. The data is interpreted correctly and consistently throughout the entire manuscript. The statements and conclusions formulated are generally consistent and are supported by the listed sources and research results. The figures/tables/images/diagrams are presented correctly.
The recommendations for further work are as follows. It would be interesting in future studies to increase the sample size in the SLE group to increase statistical power, and we would also like to suggest that the authors consider conducting longitudinal studies to establish cause-effect relationships and study the dynamics of changes. Overall, the article is a valuable contribution to the study of the psychological effects of exercise on older women. Despite some shortcomings, the study is well planned, conducted and analyzed. The results may be useful for developing effective physical activity programs aimed at improving the mental health of the elderly.
Author Response
Response to Editor and Reviewers
August 8, 2025
Dear Editor and Anonymous Reviewers,
Greetings, thank you for your valuable feedback on our manuscript (Submission: brainsci-3779781). We have carefully revised the manuscript in accordance with the constructive suggestions of the reviewers. We are confident that these revisions have significantly enhanced the quality and value of our manuscript. Please note that the reviewer's comments will be displayed in red, our responses will appear in blue, and information from the revised document will be highlighted in green. The specifics of the revisions are as follows:
Reviewer 3
Comment 1:
The structure of the article is generally logical, but there are some areas that can be improved for greater clarity and consistency. The manuscript is generally understandable, although some sections, especially in the discussion, could have been more concise and focused. The topic is relevant for the field of gerontology and public health. The structure of the article corresponds to the standard format of scientific publications (introduction, methods, results, discussion, conclusions).
Response: We sincerely thank the reviewer for the positive feedback regarding the overall structure, relevance, and clarity of our manuscript, as well as for the constructive suggestions to improve certain sections. We agree that enhancing clarity, ensuring consistency, and making the discussion more concise and focused will strengthen the manuscript. In response, we have carefully revised the identified sections, particularly in the Discussion, to improve focus and remove redundancies. We have also made major adjustments throughout the manuscript to ensure consistency in style and presentation. The revised portions are highlighted in yellow for your convenience.
Comment 2:
However, the introduction could be made shorter and the discussion more structured, with a clear division into sub–topics. The cited sources are fresh and meet the requirements of the journal. The manuscript is scientifically based, the authors rely on the existing literature and theoretical framework. The experimental design (cross-sectional study) is suitable for testing hypotheses, although it has limitations in establishing causal relationships. The sample (older women in China) is justified, given the cultural characteristics and the growing proportion of the elderly population in this country. The Methods section contains enough details to reproduce the research results. The values of Cronbach's alpha for the scales used are indicated, which indicates the reliability of the instruments. The data is interpreted correctly and consistently throughout the entire manuscript. The statements and conclusions formulated are generally consistent and are supported by the listed sources and research results. The figures/tables/images/diagrams are presented correctly.
Response: Thank you for your detailed and constructive feedback on our manuscript. We have made improvements to the introduction and discussion sections based on your suggestions. Please see pages 2-3, Line 45-94, Introduction. Please see pages 13-15, Line 345-447, Discussion.
Comment 3:
The recommendations for further work are as follows. It would be interesting in future studies to increase the sample size in the SLE group to increase statistical power, and we would also like to suggest that the authors consider conducting longitudinal studies to establish cause-effect relationships and study the dynamics of changes. Overall, the article is a valuable contribution to the study of the psychological effects of exercise on older women. Despite some shortcomings, the study is well planned, conducted and analyzed. The results may be useful for developing effective physical activity programs aimed at improving the mental health of the elderly.
Response: Thank you for your positive evaluation of our research and your valuable suggestions for the subsequent work. We fully agree with your suggestions for future research, we added the future research direction in the limitation section...
“Future research should focus on four directions: 1. Expanding the SLE sample via stratified sampling and applying rigorous statistical models to enhance generalizability.” Please see pages 15, Line 441-442.
We would like to sincerely thank our editor and reviewers for your invaluable advice and comments, which greatly enhanced and streamlined our manuscript. Should any further inquiries arise, please feel free to reach out; we are more than willing to engage in additional discussions to ensure the continued improvement of our work.

Round 2
Reviewer 1 Report
Comments and Suggestions for Authors
After analysis, although the manuscript does not present a theme capable of generating major advances, the present manuscript is in better conditions for approval.